# WHERE TO GO NEXT FOR RECOMMENDER SYSTEMS? ID- VS. MODALITY-BASED RECOMMENDER MODELS REVISITED

## ABSTRACT

Recommender models that utilize unique identities (IDs for short) to represent distinct users and items have been the state-of-the-arts and dominating the recommender system (RS) literature for over a decade. In parallel, the pre-trained modality encoders, such as BERT (Devlin et al., 2018) and ResNet (He et al., 2016), are becoming increasingly powerful in modeling raw modality features, e.g., text and images. In light of this, a natural question arises: whether the modality (a.k.a, content) only based recommender models (MoRec) can exceed or be on par with the ID-only based models (IDRec) when item modality features are available? In fact, this question had been answered once a decade ago, when IDRec beat MoRec with strong advantages in terms of both recommendation accuracy and efficiency.

We aim to revisit this 'old' question and systematically study MoRec from several aspects. Specifically, we study several sub-questions: (i) which recommender paradigm, MoRec or IDRec, performs best in various practical scenarios, including regular, cold and new item scenarios? does this hold for items with different modality features? (ii) will MoRec benefit from the latest technical advances in corresponding communities, for example, natural language processing and computer vision? (iii) what is an effective way to leverage item modality representations, freezing them or adapting them by fine-tuning on new data? (iv) are there any other factors that affect the efficacy of MoRec. To answer these questions, we conduct rigorous experiments for item recommendations with two popular modalities, i.e., text and vision. We provide empirical evidence that MoRec with standard end-to-end training is highly competitive and even exceeds IDRec in some cases. Many of our observations imply that the dominance of IDRec in terms of recommendation accuracy does not hold well when items' raw modality features are available. We promise to release all related codes & datasets upon acceptance.

## 1 INTRODUCTION

Recommender systems (RS) model the historical interactions of users and items and recommend items that users may interact with in the future. RS are playing a key role in search engines, advertising systems, e-commerce websites, video and music streaming services, and various other Internet platforms. Mainstream recommender models usually use unique IDs to represent items, which can be broadly categorized into two classes: two-tower based architectures (Rendle et al., 2012; Huang et al., 2013) and sequence or session-based neural architectures (Hidasi et al., 2015; Yuan et al., 2019; Kang & McAuley, 2018; Sun et al., 2019). These ID-only or ID-based recommender models (IDRec) are well-established and have been dominating the RS field for over a decade.

Despite their popularity and success, there are also key weaknesses that should not be ignored. First, IDRec highly rely on the ID interactions, which fail to provide recommendations when users and items have few interactions (Yuan et al., 2020), a.k.a. the cold-start setting. Second, pre-trained IDRec are not transferable across platforms given that user IDs and item IDs are in general not shareable in practice. This issue seriously limits the development of big & general-purpose RS models (Ding et al., 2021; Bommasani et al., 2021; Wang et al., 2022), an emerging paradigm in other deep learning application areas. Third, IDRec represent items mainly by ID embedding

features, ignoring the inherent content features and thus are prone to achieving sub-optimal performance. Moreover, maintaining a large and frequently updated ID embedding matrix for users and items remains a key challenge in industrial applications (Sun et al., 2020). Beyond these issues, ID-only recommender models cannot benefit from advances in other communities, such as powerful representation models developed in NLP (natural language processing) and CV (computer vision) areas. Last but not the least, recommender models leveraging ID features have obvious drawbacks in terms of interpretability, visualization and evaluation.

In contrast to IDRec, content-based recommender models (CoRec) rely heavily on item features, i.e., characteristics of the item such as the color of an object, authors of a book, and keywords in an article. While intuitive and interpretable, they have been far less prevalent than IDRec over the past decade. A key reason for this could be that the content-based item encoders are not as expressive as the standard item ID embedding, therefore leading to unsatisfactory performance. Nevertheless, we believe that given the recent extraordinary success of deep representation learning, it is time to revisit the critical comparison between CoRec and IDRec. In particular, BERT (Devlin et al., 2018), GPT-3 (Brown et al., 2020) and Vision Transformers (Dosovitskiy et al., 2020; Liu et al., 2021) have revolutionized the NLP and CV fields in terms of representing the raw text and vision features. Whether the item representations learned by these backbone models are better suited for recommender systems than ID embeddings remains largely unknown until now.

In this paper, we intend to rethink the potential of CoRec and study a key question: *should we still stick to the ID-based recommender paradigm?* We concentrate on item recommendation based on the text and vision modalities — the two most common modalities in literature. To differentiate from traditional attribute-based CoRec, we refer to recommender models directly encoding items' raw modality features as MoRec. To be concise, we attempt to address the following sub-questions:

**Q(i): Equipped with strong modality encoders (ME), can MoRec perform comparably or better than IDRec in various recommendation scenarios?** To answer this question, we conduct empirical studies by taking into account the **two** most representative recommender architectures (i.e., two-tower based DSSM (Huang et al., 2013; Rendle et al., 2020) and session-based SASRec (Kang & McAuley, 2018)) equipped with **four** powerful ME evaluated on **three** large-scale recommendation datasets with **two** modalities (text and vision) and **three** recommendation scenarios (regular, cold & new item settings).

**Q(ii): If Q(i) is yes, can the recent technical advances developed in NLP and CV fields be translated into accuracy improvement for MoRec when they utilize text and vision features?** We address this question by performing three experiments. First, we evaluate MoRec by comparing modality-based item encoders (e.g. BERT and ResNet (He et al., 2016)) with *vs* without pre-training on corresponding NLP and CV datasets; second, we evaluate MoRec by comparing weaker *vs* stronger ME where weaker and stronger are determined by NLP and CV tasks; third, we evaluate MoRec by comparing smaller *vs* larger ME given that ME with larger model sizes tend to perform better than their smaller counterparts in various downstream tasks.

**Q(iii): How can we effectively employ item modality representations derived from an NLP or CV encoder network? Is the end-to-end (E2E) fine-tuned representation largely superior to the frozen representation given that the E2E training fashion requires much more compute and training time?** The de facto practice for industrial recommender systems is to first extract item modality representations through some ME as 'off-the-shelf' features and then incorporate them into a recommender model (McAuley et al., 2015; Covington et al., 2016), often referred to as the two-stage (TS) paradigm. While such TS paradigm is architecturally flexible, easy-to-implement and requires less compute and training time, we show that there is a substantial accuracy loss compared to the E2E paradigm.

Beyond these key questions, we also identify several other factors that affect the training of MoRec in practice. To serve as a foundation for further research of MoRec, we will publish all our codes and datasets, including a large-scale real-world video recommendation dataset (collected by ourselves) containing over 4 million user-video interactions with around $128K$ video thumbnails and $400K$ users.[1]

---

[1] $K$ is short for thousand.

## 2 IDREC & MOREC

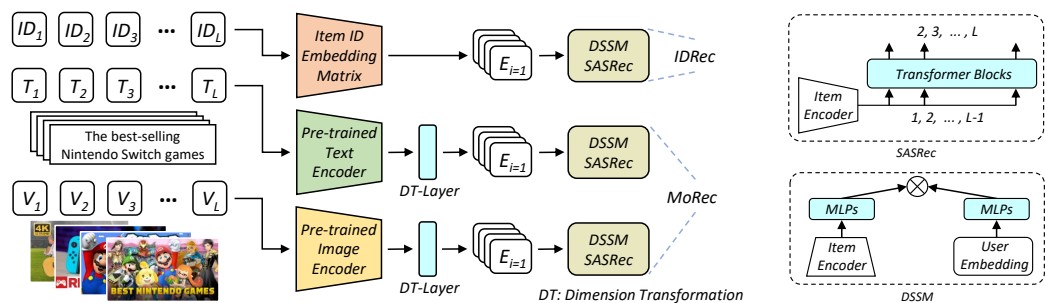

Figure 1: Illustration of IDRec vs MoRec. $V_i$ and $T_i$ denote raw features of vision and text modalities. $E_i$ is the item representation vector fed into the recommender model. The only difference between IDRec and MoRec is the item encoder. IDRec use an item ID embedding matrix as the item encoder, whereas MoRec use the pre-trained ME (followed by a dense layer for the dimension transformation, denoted by DT-layer) as the item encoder.

One core function of a recommender model is to represent items and users. Denote $\mathcal{I}$ (of size $|\mathcal{I}|$) and $\mathcal{U}$ (of size $|\mathcal{U}|$) as the set of items and users respectively. For an item $i \in \mathcal{I}$, we can represent it either by its unique ID $i$ or its modality content, such as text and image features. Likewise, for a user $u \in \mathcal{U}$, we can represent her either by the unique ID $u$ or the profile of $u$, where a profile can be the demographic information or a sequence of interacted items.

For IDRec, an ID embedding matrix $\mathbf{X}^{\mathcal{I}} \in \mathbb{R}^{|\mathcal{I}| \times d}$ is constructed as the item encoder for all items in $\mathcal{I}$, where $d$ is the embedding size. During training and inference, IDRec retrieve $\mathbf{X}^{\mathcal{I}}_i \in \mathbb{R}^d$ from $\mathbf{X}^{\mathcal{I}}$ as the embedding of item $i$ and then feed it to the recommender network.

In MoRec, items are assumed to contain modality information beyond their IDs. For item $i$, MoRec use the modality encoder (ME) to generate the representation for the raw modality input of $i$ and use it to replace the ID embedding vector in IDRec. For instance, in the news recommendation scenario, we can use the pre-trained BERT or RoBERTa (Liu et al., 2019) as text ME and represent a piece of news by the output textual representation of its title. Similarly, when items contain visual features, we can simply use a pre-trained ResNet or Vision Transformer as vision ME.

In this paper, we perform rigorous empirical studies on the two most commonly adopted recommender paradigms: DSSM (Huang et al., 2013) and SASRec (Kang & McAuley, 2018). The DSSM model is a two-tower based architecture where users/items are encoded by their own encoder networks with user and item IDs as input. SASRec is a well-known sequential recommender model based on multi-head self-attention (Vaswani et al., 2017) which describes a user by her interacted item ID sequence. As mentioned before, by replacing the ID embedding matrix with an item modality encoder, we obtain the MoRec version of both DSSM and SASRec. We illustrate IDRec and MoRec in Figure 1, and provide training details in Appendix A.

## 3 EXPERIMENTAL SETUPS

### 3.1 DATASETS

We evaluate IDRec and MoRec on three real-world datasets, namely, the MIND dataset from the Microsoft news recommendation platform (Wu et al., 2020), the HM clothing purchase dataset from the H&M platform[2] and the Bili dataset (see Appendix C for data collection details) from an online video recommendation platform[3]. For MIND, we use the title of each article to represent the item, while for HM & Bili, we represent items by their associated images (one image per item). To fully exploit the capabilities of MoRec, the dataset used should ensure that the user's decisions

---

[2]https://www.kaggle.com/competitions/h-and-m-personalized-fashion-recommendations/overview
[3]https://www.bilibili.com/

(whether to interact with an item or not) are determined only by the content features of the item. We notice that neither HM nor Bili meets this assumption since, in addition to the appearance of the item, users' purchase decisions on HM are also influenced by the price and textual description of the product, while users' commenting behavior on Bili is mainly affected by the video (with hundreds of images) and audio signals (not only the cover image).(see Appendix C Figure 6). That is, **HM and Bili are not well suited for MoRec when using only an image to represent the item.**[4]

Table 1: Dataset characteristics. $n$ and $m$ denote the numbers of users and items respectively. $|\mathcal{R}|^{train}$, $|\mathcal{R}^{valid}|$, $|\mathcal{R}^{test}|$, $|\mathcal{R}^{cold}|$ and $|\mathcal{R}^{new}|$ denote the number of interactions of the training set, validation set, testing set, cold items, and new items respectively. $|\mathcal{R}|/(nm)$ represents density.

| Dataset | $n$ | $m$ | $|\mathcal{R}|^{train}$ | $|\mathcal{R}^{valid}|$ | $|\mathcal{R}^{test}|$ | $m^{cold}$ | $m^{new}$ | $|\mathcal{R}^{train}|/(nm)$ | Behavior Types |
|---------|-----|-----|------------------------|------------------------|-----------------------|-----------|-----------|-----------------------------|----------------|
| MIND | $630K$ | 79,707 | $8,407K$ | $630K$ | $630K$ | 32,246 | 13,133 | 0.0167% | clicks |
| HM | $500K$ | 86,733 | $5,500K$ | $500K$ | $500K$ | 37,087 | 14,498 | 0.0127% | purchases |
| Bili | $400K$ | 127,625 | $4,400K$ | $400K$ | $400K$ | 39,331 | 5,030 | 0.0086% | comments |

To construct the datasets, we randomly select around $400K$, $500K$ and $600K$ users from Bili, HM, and MIND, respectively. Then, we perform basic pre-processing by setting the size of all images to $224 \times 224$ and the title of all news articles to a maximum of 30 tokens (covering 99% of descriptions). For MIND, we select the latest 23 items for each user to construct the interaction sequence. For HM and Bili, we choose the 13 most recent interactions since encoding images requires much larger GPU memory (especially with the SASRec architecture). Following (Rendle et al., 2012), we remove users with less than 5 interactions, simply because we do not consider cold user settings in this paper. Moreover, we do consider the cold item setting. Specifically, we count the interactions of all items in the training set (data split is described below) and regard those that appear less than 10 times as cold items and those that never appear as new items. Then we select additional user sequences out of the training dataset that contain such cold or new items to perform evaluation.[5] We report the statistics of all processed datasets in Table 1.

## 3.2 EVALUATIONS

We split the datasets into training, validation, and testing sets by adopting the standard leave-one-out strategy. Specifically, the latest interaction of each user was used for evaluation, while second-to-last was used as validation for hyper-parameter searching, and all others are used for training. We evaluate all models using two popular top-N ranking metrics: HR@N (Hit Ratio) and NDCG@N (Normalized Discounted Cumulative Gain) (Yuan et al., 2019), where N is set to 10. We rank the ground-truth target item by comparing it with all the other items in the item pool. Finally, we report the results on the testing set by choosing checkpoints with the best validation performance.

## 3.3 COMPARISON SETTINGS

For a fair comparison, we ensure that IDRec and MoRec have exactly the same network architecture except the item encoder. For both text and vision encoders, we pass their output item representations to a DT-layer (see Figure 1) for dimension transformation. Regarding the hyper-parameter setting, our principle is to ensure that IDRec are always thoroughly tuned, including learning rate $\gamma$, embedding size $d$, layer number $l$, dropout $\rho$, etc. While for MoRec, we attempt to first use the same set of hyper-parameters as IDRec and then perform some basic searching around the best choices. Thereby, without specially mentioning we do not guarantee that MoRec are reported with the best results as searching all possible hyper-parameters for MoRec is super expensive and time-consuming. We report detailed settings in the Appendix G Table 12.

## 4 COMPARATIVE STUDIES (Q(I))

We perform evaluation on three RS scenarios, namely, the canonical scenario (CANO-SC) with both warm, cold and new items, cold item scenario (COLD-SC) and new item scenario (NEW-SC).

---

[4]Nevertheless, there is no ideal publicly available image recommendation dataset (with raw image features) where a user's interaction decisions are only or mostly determined by the image itself.

[5]This is simply because the original testing set has too few cold items for evaluation.

Table 2: Accuracy (%) comparison of IDRec and MoRec using DSSM and SASRec in CANO-SC. MoRec with different ME are directly denoted by their encoder names for clarity. The best results for DSSM and SASRec are bolded. 'Improv.' is the relative improvement of the best MoRec compared with the best IDRec. All results of MoRec are obtained by fine-tuning their whole parameters including both the item encoder and user encoder. Swin-T and Swin-B are Swin Transformer with different model sizes, where T is tiny and B is base. ResNet50 is a 50-layer ResNet variant.

| Dataset | Metrics | DSSM | | | SASRec | | | | Improv. |
|---------|---------|------|---|---|--------|---|---|---|---------|
| | | IDRec | BERT$_{base}$ | RoBERTa$_{base}$ | IDRec | BERT$_{small}$ | BERT$_{base}$ | RoBERTa$_{base}$ | |
| MIND | HR@10 | **3.58** | 2.68 | 3.07 | 17.71 | 18.50 | 18.23 | **18.68** | +5.48% |
| | NDCG@10 | **1.69** | 1.21 | 1.35 | 9.52 | 9.94 | 9.73 | **10.02** | +5.25% |
| | | IDRec | ResNet50 | Swin-T | IDRec | ResNet50 | Swin-T | Swin-B | |
| HM | HR@10 | **4.93** | 1.49 | 1.87 | 6.84 | 6.67 | 6.97 | **7.24** | +5.85% |
| | NDCG@10 | **2.93** | 0.75 | 0.94 | **4.01** | 3.56 | 3.80 | 3.98 | -0.75% |
| Bili | HR@10 | **1.14** | 0.38 | 0.57 | 3.03 | 2.93 | 3.18 | **3.28** | +8.25% |
| | NDCG@10 | **0.56** | 0.18 | 0.27 | 1.63 | 1.45 | 1.59 | **1.66** | +1.84% |

## 4.1 MOREC VS IDREC ON CANO-SC

Here, we evaluate IDRec and MoRec with the two most important recommender architectures, i.e., DSSM and SASRec. We use pre-trained BERT and RoBERTa as ME when items are of text features, and use pre-trained ResNet and Swin Transformer (Liu et al., 2021) when items are of visual features. We provide details (model size & download urls) of these ME in Appendix G Table 9. Note for BERT and RoBERTa, we add the DT-layer on the final representation of the "[CLS]" token. We report results on the testing set in Table 2 and convergence curves on Appendix B Figure 4 (and Appendix F.2 Figure 10 with session length of 23 for vision recommendations).

First, we observe that DSSM always substantially underperforms SASRec, regardless of the item encoding strategy used. For instance, SASRec-based IDRec is around $4.9\times$ better than DSSM-based IDRec in terms of HR@10 for news recommendation, although their training, validation, and testing sets are kept the same. The performance gap for image recommendation is relatively small, around $1.4\times$ and $2.7\times$, on HM and Bili respectively. This is expected since much prior literature (Kang & McAuley, 2018; Sun et al., 2019) has revealed that properly representing users with their interacted item sequence — e.g., by an autoregressive training manner (Kang & McAuley, 2018; Yuan et al., 2019) — is in general more powerful than dealing them as individual user IDs. Besides, DSSM neither explicitly, nor implicitly, takes into account interaction orders which reflect user's dynamic preference, thereby leading to inferior results for time-aware item recommendations.

Second, we notice that with the DSSM architecture, MoRec perform much worse than IDRec in all three datasets even with the SOTA ME, in particular for the visual recommendation scenarios. By contrast, with the SASRec architecture, MoRec consistently achieve better results than IDRec on MIND using any of the three text encoders, i.e., BERT$_{small}$, BERT$_{base}$ and RoBERTa$_{base}$. For instance, MoRec outperform IDRec by over 5% on the two evaluation metrics with the RoBERTa$_{base}$ text encoder. Meanwhile, MoRec perform comparably to IDRec when using Swin Transformer as ME but perform relatively worse when using ResNet50. The performance disparity of MoRec between DSSM and SASRec potentially implies that **a powerful recommender architecture (SAS-Rec *vs* DSSM) is required to fully harness the strengths of the modality-based item encoder**.[6]

## 4.2 MOREC VS IDREC ON COLD-SC & NEW-SC

MoRec are a natural fit for cold item recommendation as their ME module is specifically developed to model the raw modality features of an item, whether it is cold or not. To validate this, we evaluate IDRec and MoRec in the two scenarios, i.e., COLD-SC and NEW-SC. We report the results in Appendix D Table 7. As clearly shown in Table 7, MoRec consistently and substantially improve IDRec on all three datasets for both text and vision modalities in both COLD-SC and NEW-SC. In particular, the HR@10 on MIND rises from 0.0036% to 3.0637% in COLD-SC and from 0.0125%

---

[6]Given MoRec's poor results with DSSM, we only focus on the SASRec architecture in the following.

to 0.5899% in NEW-SC.[7] Similar observations can also be made for the image and video recommendation on HM and Bili. The superiority of MoRec comes from the powerful representations of ME which is first pre-trained on large-scale text and image datasets and then fine-tuned to adapt the recommendation objectives.

The above results shed the following insights: (1) the recommender architecture matters a lot for the performance of MoRec; (2) the item encoder network influences the performance of MoRec as well; (3) (**Answer for Q(i)**) **equipped with the most powerful ME, MoRec clearly beat the IDRec counterpart for text recommendation and is on par with IDRec for vision recommendation with the sequential recommender architecture. However, it seems that there is little chance for MoRec to replace IDRec with the typical two-tower DSSM paradigm in the canonical scenario, in particular for items with vision modality.** (4) Without any doubt, MoRec markedly outperform IDRec in the cold-start and new item settings. It is worth mentioning again that *Bili and HM datasets are somewhat unfair for MoRec since their interacted items are not mostly determined by image features (e.g., it is nearly impossible to learn the price or audio features from only a raw image).* Suggested by reviewr jqDE & tNVZ: We also present the cross-domain recommendation task of MoRec in Appendix J.

## 5 CAN MoRec inherit advances in multimedia communities? (Q(ii))

One key advantage of MoRec is that it promises to introduce the strong representation learning techniques from other communities, e.g., NLP and CV, to the recommendation task. Hence, we ask: can the latest advances in NLP and CV be transferred to the tasks of recommender systems? We aim to answer this question from the following perspectives.

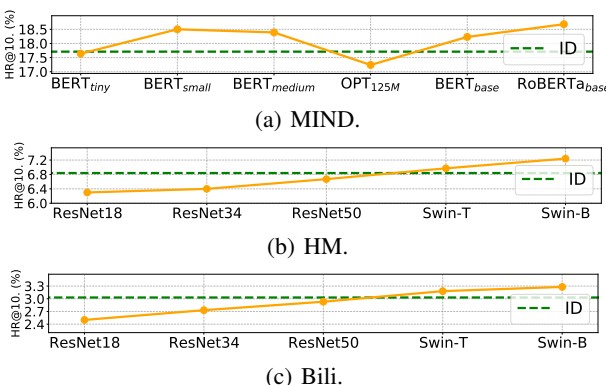

(a) MIND.

(b) HM.

(c) Bili.

Figure 2: Accuracy with different pre-trained ME in MoRec. Parameters of the pre-trained encoder network are all fine-tuned on the recommendation task.

Table 3: Pre-trained (PE) ME vs TFS on the testing set regarding HR@10 (%). $\text{BERT}_{base}$ are used as text ME, and ResNet50 and Swin-T are used as vision ME. 'Improv.' indicates the relative improvement of PE over TFS. We report the convergence behaviors on the validation set in Appendix E Figure 7.

| Dataset | Encoder | TFS | PE | Improv. |
|---------|---------|-----|-----|---------|
| MIND | $\text{BERT}_{base}$ | 17.78 | 18.23 | +2.53% |
| HM | ResNet50 | 5.82 | 6.67 | +14.60% |
| | Swin-T | 6.27 | 6.97 | +11.16% |
| Bili | ResNet50 | 2.67 | 2.93 | +9.74% |
| | Swin-T | 2.83 | 3.18 | +12.37% |

First, we investigate whether a larger pre-trained ME enables better recommendation accuracy since in NLP and CV larger pre-trained models tend to offer higher performance in corresponding downstream tasks. As shown in Figure 2, a larger vision item encoder always achieves better image recommendation accuracy, i.e., ResNet18-based MoRec < ResNet34-based MoRec < ResNet50-based MoRec, and Swin-T based MoRec < Swin-B based MoRec. Similarly, we find that $\text{BERT}_{tiny}$-based MoRec < $\text{BERT}_{small}$-, $\text{BERT}_{medium}$-, $\text{BERT}_{base}$-based MoRec. One difference is that $\text{BERT}_{medium}$- and $\text{BERT}_{base}$-based MoRec do not outperform $\text{BERT}_{small}$-based MoRec although the latter has a smaller-size BERT variant. We conclude that in general larger ME tend to improve the recommendation accuracy but this may not strictly hold in all cases.

Second, we investigate whether a more powerful encoder network enables better recommendations (see Appendix G Table 9 for more details about these ME). For example, it is recognized that RoBERTa outperforms BERT (Liu et al., 2019), and BERT outperforms the unidirectional

---

[7]We find that the accuracy of IDRec on MIND in COLD-SC is worse than the random strategy (HR@$N \approx \frac{1}{m} \times N$) used for new item recommendation (e.g., 0.0036 vs. 0.0125). This is reasonable since cold items have fewer chances to be ranked higher after training IDRec, sometimes even worse than a basic random sampler.

GPT (Radford et al., 2018), e.g., OPT, for most text understanding (but not generative) tasks with similar model size, and that Swin Transformer often outperforms ResNet in many CV tasks (Liu et al., 2021). As shown in Figure 2, MoRec's performance keeps consistent with the findings in NLP and CV, i.e., RoBERTa$_{base}$-based MoRec > BERT$_{base}$-based MoRec > OPT$_{125M}$-based MoRec, and Swin-T based MoRec > ResNet50-based MoRec (Swin-T has a similar model size to ResNet50, the same for RoBERTa$_{base}$, BERT$_{base}$ and OPT$_{125M}$).

Third, we investigate whether the pre-trained ME produces higher recommendation accuracy than its training-from-scratch (TFS) version (i.e., with random initialization). There is no doubt that the pre-trained BERT, ResNet, and Swin largely improve corresponding NLP and CV tasks against their TFS versions. We report the recommendation results on the testing set in Table 3 and the convergence behaviors on the validation set in Appendix E Figure 7. It can be clearly seen that pre-trained MoRec obtain better convergence and final results. In particular, MoRec achieve around 10% improvements with the pre-trained ME (ResNet and Swin) on HM and Bili, which also aligns with findings in NLP and CV domains.

According to the above experiments, we conclude that **(Answer for Q(ii)) MoRec build connections for RS and other multimedia communities, and can in general inherit the latest advances from the NLP and CV fields**. In other words, MoRec have more chances to be improved in the future as long as new breakthrough happens in corresponding research fields.

# 6 END-TO-END (E2E) VS TWO-STAGE (TS) TRAINING (Q(III))

E2E-based MoRec incur a significant computational overhead due to forward and backward propagation for each item in a given sequence of user interactions. The common approach in practice is usually based on a TS manner: first extracting offline features by ME and then adding them into a recommender network (He & McAuley, 2016b;a) (see Figure 3). Such TS pipeline is especially popular for industrial applications (e.g., Deep Crossing (Shan et al., 2016) and YouTubeDNN (Covington et al., 2016)) considering that there are often hundreds of millions of training examples. Here, we evaluate whether this widely adopted strategy with fixed features leads to ideal recommendation.

As shown in Table 4, we find that TS-based MoRec show surprisingly poor results, compared to IDRec and E2E-based MoRec. In particular, with ResNet, it achieves only 60% and 25% performance of E2E-based MoRec on HM and Bili respectively. The results indicate that the modality features pre-trained by these NLP and CV tasks are not universal enough, which thereby results in inferior recommendation results. MoRec achieve the best results by performing further parameter fine-tuning, which needs the E2E training manner (Suggested by reviewer jqDE: For a comprehensive comparison, we also add several additional neural network layers on the top of the extracted item representation so as to see whether such strategy will better align these fixed representation with the current recommender backbone network. We report the results in Appendix I). Thereby, we want to remind researchers and practitioners that **(Answer for Q(iii)) the de facto two-stage recommender regime causes severe performance degradation, which should not be ignored in practice.** Meanwhile, we advocate more efforts on optimizing the training[8] cost and efficiency of the E2E-based MoRec since it cause around 100x-1000x large compute & training time than IDRec (see Appendix Table 12).

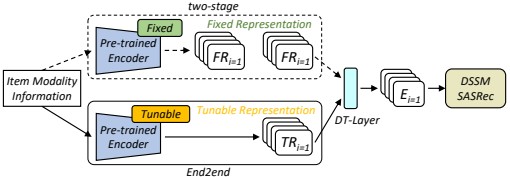

Figure 3: Illustration of MoRec with TS and E2E.

Table 4: E2E vs TS in terms of HR@10 (%).

| Dataset | IDRec | Encoder | TS | E2E |
|---------|-------|---------|-----|-----|
| MIND | 17.71 | BERT$_{base}$ | 13.93 | 18.23 |
| HM | 6.84 | ResNet50 | 4.03 | 6.67 |
| | | Swin-T | 3.45 | 6.97 |
| Bili | 3.03 | ResNet50 | 0.72 | 2.93 |
| | | Swin-T | 0.79 | 3.18 |

---

[8]For the online inference stage, MoRec are as fast as IDRec since ME can be calculated by offline manner.

## 7    BAG OF OTHER OBSERVATIONS

E2E-based MoRec have been less studied before, especially for visual recommendation. Here, we present several other findings (including both positive and negative results) towards practical MoRec.

**A second round of pre-training on ME.** Performing a second round of pre-training for ME using the downstream dataset (without using labels) sometimes works well in the DL literature (Gururangan et al., 2020). Here, we explore whether it offers improved recommendations for MoRec. Following the pre-training of BERT, we adopt the "masked language model" (MLM) objective to train the text encoder of MoRec (denoted by $BERT_{base}$-MLM) on MIND and report results in Table 5. As shown, $BERT_{base}$-MLM gains higher accuracy than $BERT_{base}$ for both the TS and E2E models. Similarly, we explore whether it holds for the vision encoder. Note that ResNet and Swin Transformer used in previous experiments are pre-trained in a supervised manner, but neither HM nor Bili contains supervised image labels. To this end, we turn to use MAE (He et al., 2022), a SOTA image encoder pre-trained in an unsupervised manner, similar to MLM. We find $MAE_{base}$-MLM clearly improves the standard $MAE_{base}$ on HM with the TS model, but obtains marginal gains with the E2E model. By contrast, no accuracy improvements are observed on Bili. By examining image cases in Appendix C Figure 5, we find that pictures in Bili have very diverse topics and are more challenging than HM (with only very simple fashion elements). Our conclusion is that **the effectiveness of the second round of pre-training depends on individual datasets; more importantly, it seems very difficult to obtain larger accuracy gains for the E2E strategy.**

**Trade-off between accuracy and efficiency.** While fine-tuning full parameters in ME produces better accuracy than frozen features, it also imposes a greater computational burden. Here, we attempt to investigate whether all parameters need adaptation. As shown in Table 6, MoRec with BERT-based ME show the optimal results when fine-tuning the top 6 blocks (approximately a half of all parameters). For the two visual datasets, fine-tuning all layers yields slightly better results than fine-tuning the top 2 out of 4 blocks. According to this, we conclude that **in practice, fine-tuning the top half of ME is worth considering for the balance between accuracy and efficiency.** [9]

Besides, we have examined separate parameter searching for item & user encoders, and report the results in Appendix H. Our conclusion is **searching hyperparameters (e.g., learning rate & weight decay) separately for ME and user encoder matters for obtaining better MoRec results.**

Table 5: Comparison of HR@10 (%) w/ and w/o extra pre-training with the MLM objective using the TS and E2E training strategy. 'Improv.' means the relative improvement of w/ MLM compared to w/o MLM.

| Dataset | Encoder | Manner | w/o MLM | w/ MLM | Improv. |
|---------|---------|--------|---------|--------|---------|
| MIND | $BERT_{base}$ | TS | 13.93 | 14.68 | +5.38% |
|  |  | E2E | 18.23 | 18.63 | +2.19% |
| HM | $MAE_{base}$ | TS | 2.50 | 2.79 | +11.60% |
|  |  | E2E | 7.03 | 7.07 | +0.57% |
| Bili | $MAE_{base}$ | TS | 0.57 | 0.57 | 0.00% |
|  |  | E2E | 3.18 | 3.17 | -0.31% |

Table 6: Comparison of HR@10 (%) when fine-tuning different blocks of the pre-trained ME. 'n/m' means fine-tuning the top n blocks of ME containing a total of m blocks.

| Dataset | Encoder | Fine-tuning blocks | | | |
|---------|---------|------|------|------|------|
|  |  | 0/12 | 2/12 | 6/12 | 12/12 |
| MIND | $BERT_{base}$ | 13.93 | 17.87 | **18.51** | 18.23 |
|  |  | 0/4 | 1/4 | 2/4 | 4/4 |
| HM | ResNet50 | 4.03 | 6.46 | 6.59 | **6.67** |
|  | Swin-T | 3.45 | 6.18 | 6.80 | **6.97** |
| Bili | ResNet50 | 0.72 | 2.83 | 2.89 | **2.93** |
|  | Swin-T | 0.79 | 2.88 | 3.00 | **3.18** |

## 8    RELATED WORK

**ID-based recommender systems (IDRec).** There are thousands of recommender models purely based on user/item ID as inputs, ranging from early item-to-item collaborative filtering (Linden et al., 2003), latent factorization models (Koren et al., 2009; Rendle, 2010), to deep learning (DL) models (He et al., 2017; Hidasi et al., 2015). According to whether explicitly modeling users' temporal preferences, they can be roughly divided into two types: two-tower based models (TTRM) and

---

[9]We have not reported such results since it can be easily analyzed from Appendix G Table 12 that both training time, compute resources and GPU memory consumption will be largely reduced by fine-tuning only a few layers.

sequential neural models (SRM). TTRM (Rendle et al., 2012; Huang et al., 2013; He & McAuley, 2016b; Covington et al., 2016) assign a unique user ID to represent the user and posses both user encoder and item encoder. Despite the remarkable success of DL, TTRM in literature still has very shallow layers (Yang et al., 2020; Rendle et al., 2020). By contrast, SRM uses the user-item interaction sequence to represent a user, which can be stacked with deeper layers (Yuan et al., 2019; Wang et al., 2021; Sun et al., 2020) and is typically more powerful than TTRM. The most representative sequential RM include GRU4Rec (Hidasi et al., 2015), NextItNet (Yuan et al., 2019), SASRec and BERT4Rec (Sun et al., 2019) with RNN, CNN, Transformer and BERT as the backbone respectively, among which SASRec often performs the best in the literature (Fischer et al., 2021).

**Modality-based recommender systems (MoRec).** MoRec, falls into the direction of content-based RS, and focuses on modeling item's modality features, such as text (Wu et al., 2020), images (McAuley et al., 2015), videos (Deldjoo et al., 2016), voice (Van den Oord et al., 2013) and text-image pairs (Wu et al., 2021b). Previous work tended to adopt the two-stage regime by first extracting item modality features by ME and then incorporate these fixed features into the recommender model (McAuley et al., 2015; He & McAuley, 2016b;a; Shan et al., 2016; Lee & Abu-El-Haija, 2017; Tang et al., 2019; Wei et al., 2019). More importantly, most of these works only used modality information as side features but with IDs as main features. The unpopularity of the E2E-style MoRec can be attributed into several reasons: (1) the two-stage regime is architecturally flexible for industrial applications and requires much less compute and training cost; (2) there is no high-quality publicly available dataset with raw item modality features until the recently released MIND and HM (still not ideal for such research); (3) ME in the past literature are not very expressive even by E2E training; and (4) researchers had lost confidence in content-based RS (unless for cold-start settings) since a decade ago. Some recent works started to explore the E2E-based MoRec, however, most of them focus on text recommendation. For example, Wu et al.; Shin et al.; Yu et al.; Yang et al.; Xiao et al.; Hou et al. applied different types of pre-trained ME for news recommendation; Elsayed et al. introduced ResNet to fashion-based recommendation and co-trained it with ID embeddings. However, to our knowledge, none of them performed a formal, purposeful and comprehensive study towards the comparison of IDRec and MoRec under a fair experimental setting and particularly for the non cold-start setup. A more clear comparison of related work is shown in Appendix K.

## 9 CONCLUSION, LIMITATIONS, AND FUTURE WORKS

In this paper, we studied an important yet unexplored question regarding whether IDRec would continue to dominate the RS community. Obviously, this question cannot be thoroughly addressed by one paper and need more efforts from the RS community. Yet, one major finding here is that with the SOTA ME, MoRec could already perform on par or better than IDRec with the typical recommender architecture (i.e., Transformer) even in the non-cold item recommendation setting.[10] Moreover, MoRec can largely benefit from the technical advances in the NLP and CV fields, which implies that it has larger room for accuracy improvements in the future. Given this, we believe our research is meaningful and would potentially inspire more studies on MoRec, for example, developing more powerful recommender architectures, more expressive & generalized item encoders, better item & user fusion strategies and more effective optimizations to reduce the compute & memory costs and the longer training time. We also envision that in the long run the prevailing paradigm of RS may have a chance to shift from IDRec to MoRec when raw modality features are available.

As mentioned above, this study is only a preliminary of MoRec and has several limitations: (1) we considered RS scenarios with only text and vision, whereas MoRec's behaviors with other modalities, e.g., voice and video, remain unknown; (2) we consider only single-modal item encoders, while the behaviors of multimodal MoRec are unknown; (3) we considered only a very basic approach to fusing ME into recommender models, thereby MoRec may achieve sub-optimal performance; (4) we only considered pure IDRec or MoRec, whereas the performance of a hybrid E2E-based recommender model with both ID and modality features is unknown (note in this case, some merits of MoRec will no longer hold because of the ID features); (5) it remains to be seen whether the key findings still hold if we scale up training data to $100\times$ or $1000\times$ as in many industrial systems.

---

[10]We emphasize again that since the evaluation settings are very unfair to MoRec, including unfavorable recommendation datasets (see Section 3.1), sub-optimal hyper-parameter settings (see Section 3.3), and not the largest or strongest item encoders, it might perform considerably better than IDRec on more fair settings.

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

## A    TRAINING DETAILS

Denote $\mathcal{R}$ as the set of all observed interactions in the training set. For each observed interaction $< u, i > \in \mathcal{R}$, we randomly draw a negative sample $< u, j > \notin \mathcal{R}$ in each epoch during training. Note that some literature (Yuan et al., 2016) shows that using more advanced hard negative sampling could improve the results of recommender models, which we left for future investigation. All observed interactions and sampled negative interactions can form the training set $\mathcal{R}^{train}$. Following (He et al., 2017; Kang & McAuley, 2018), we adopt the binary cross entropy loss as the objective function for both DSSM and SASRec:

$$min - \sum_{<u,i,j> \in \mathcal{R}^{sample}} \{\log(\sigma(\hat{y}_{ui})) + \log(1 - \sigma(\hat{y}_{uj}))\},\qquad(1)$$

where $\sigma(x) = 1/(1 + e^{-x})$ is the sigmoid function.

## B    CONVERGENCE OF IDREC AND MOREC ON SASREC

We show the convergence of SASRec-based IDRec and MoRec in Figure 4. Note that we report the

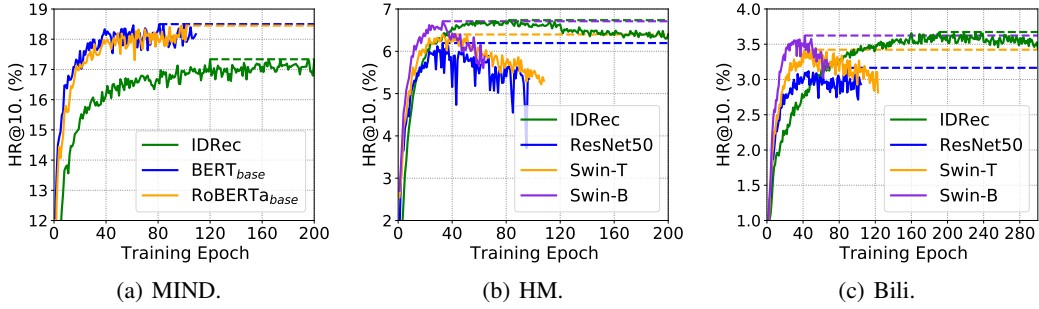

(a) MIND.                    (b) HM.                    (c) Bili.

Figure 4: Comparison of convergence of IDRec and MoRec based on the SASRec framework on the validation data of the three datasets. We only report the convergence of SASRec since it far outperforms DSSM. Note that increasing epochs does not further improve IDRec's accuracy.

results on the testing set in Table 2 by using the best validation checkpoints here. It can be seen that there is some performance gap between the validation set and testing set. In fact, we notice in much recommendation literature, authors only use training and testing set without a validation set when choosing hyperparameters — such results might be questionable.

## C    IMAGE EXAMPLES AND DATASET COLLECTION

As shown in Figure 5, we randomly pick up some image examples from the ImageNet1K dataset used for ME pre-training and the two vision datasets HM and Bili used in our experiments.

For ImageNet1K dataset in Figure 5(a), there is a wide variety of images from people's daily life and natural environments. Vision encoders such as ResNet and Swin Transformers are pre-trained on this dataset to obtain representations of arbitrary images (He et al., 2016; Liu et al., 2021).

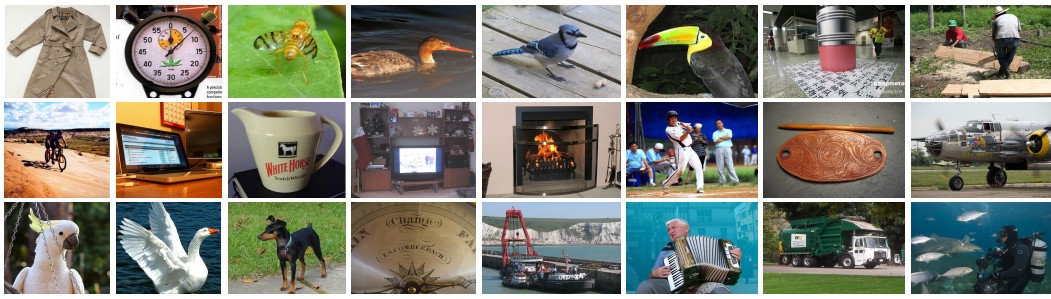

(a) ImageNet1K dataset.

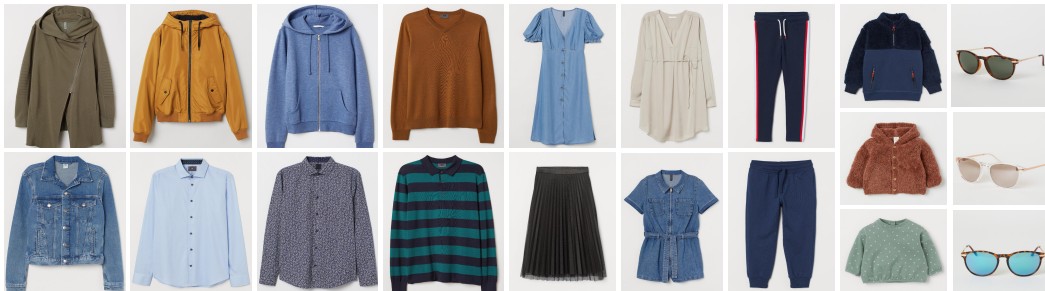

(b) HM dataset.

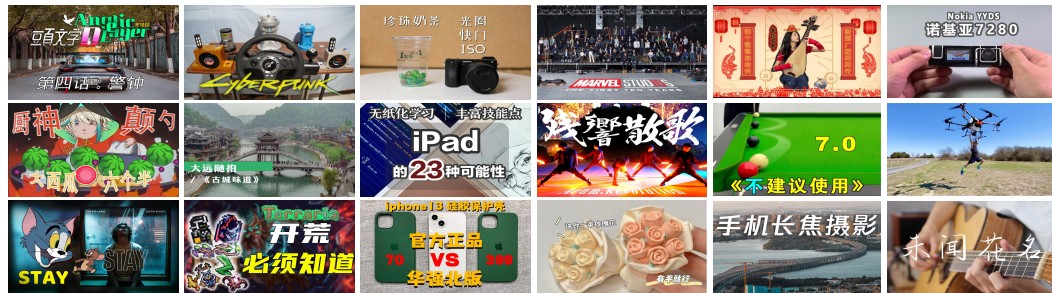

(c) Bili dataset.

Figure 5: Image examples on HM and Bili. We have also shown some examples from ImageNet, which are used to train these vision ME.

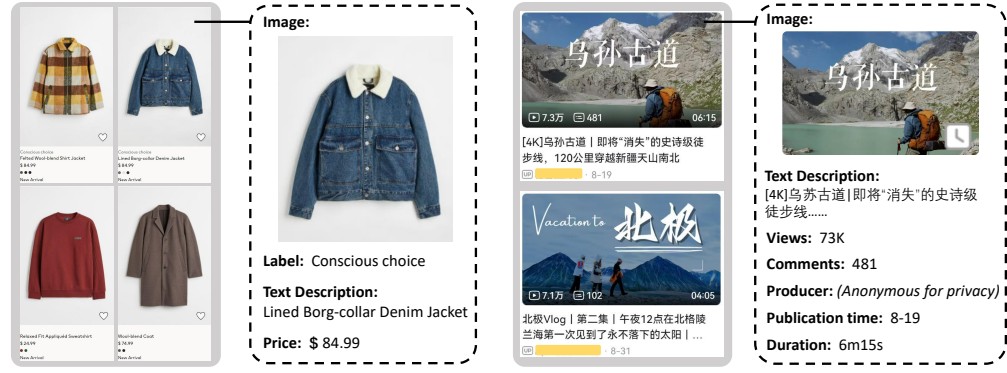

(a) Item cases on HM. Each item has a picture, a piece of text description, and price, and some items have a "conscious choice" label for sustainable material.

(b) Item cases on Bili. Every item has a thumbnail, a piece of text description, video views, video comments, producer(anonymous for privacy), publication time and video duration etc.

Figure 6: Item cases on HM and Bili datasets. We only use the raw image feature information to evaluate the performance of MoRec.

The HM dataset in Figure 5(b) contains clothing images from H&M, a clothing shopping platform. The images in HM dataset differ in the type, style and color of clothes. During the pre-training process in ImageNet1K, the vision encoder often only needs to identify a few broad categories of clothes and does not need to distinguish the details. However, clothing detail has a critical evaluation criterion for users to purchase. That might be the reason why E2E training is largely better than the two-stage freezing representations.

The Bili dataset in Figure 5(c) consists of thumbnails of videos in Bili, an online video recommendation platform. To build this dataset, **we randomly crawled videos (with duration time less than 10 minutes) from 23 different video channels in Bili and recorded their public comment information of these items as interactions. These interactions occur from 2017 to 2022. We did not crawl or use information that would involve the privacy of interacted users, whose IDs have already been anonymous (We would be very happy to provide our datasets and codes during the paper discussion period since according to the submission guidance this makes these materials only available to reviewers and AC)**. In total, we have collected over 2 million users and 150 thousand items with around 50 million interactions. However, performing research on such a large-scale dataset will require huge compute and time, thereby we simply randomly draw 400K users with their commented items as the evaluation dataset, as shown in Table 1. For this research, we only use the thumbnails to represent the item since directly encoding the original videos is super computationally expensive. As shown in Figure 5(c), the images in Bili datasets contain various domain-specific knowledge (e.g., human-created video covers with complex semantics and heavy text insertion), which may confuse the recognition of visual encoders.

We also show the actual item cases of H&M and Bili platforms in Figure 6(a) and Figure 6(b), respectively. It can be found that the actual recommendation scene contains rich information. The image and other factors (e.g., price, text description, etc.) together play a role in attracting the user's interactions. Clearly, it is impossible for MoRec to extract such features (e.g., price and audio) from only a picture — the only input of an item encoder. Note that IDRec can indeed implicitly learn such features in latent factors by item similarity supervision (Koren et al., 2009). For example, it is recognized by the RS community that ID-based location recommender models (Hang et al., 2018) can learn some good distance information even if we do not feed them distance features.

## D    MOREC VS IDREC ON COLD-SC & NEW-SC

We report the results of IDRec and MoRec on COLD-SC and NEW-SC in Table 7. Note that IDRec cannot serve new items and thereby are approximated by the simple random sampling strategy.

Table 7: HR@10 (%) of IDRec and MoRec for cold and new item recommendation. All results are evaluated based on the SASRec architecture.

| Dataset | Cold item | | New item | |
|---|---|---|---|---|
| | IDRec | BERT$_{base}$ | IDRec | BERT$_{base}$ |
| MIND | 0.0036 | 3.0637 | 0.0125 | 0.5899 |
| | IDRec | Swin-B | IDRec | Swin-B |
| HM | 0.3744 | 1.0965 | 0.0115 | 0.6846 |
| Bili | 0.3551 | 0.6400 | 0.0078 | 0.0832 |

Table 8: Pre-trained (PE) ME vs TFS (random initialization of ME) regarding HR@10 (%) in the 50K datasets.

| Dataset | Encoder | TFS | PE | Improv. |
|---|---|---|---|---|
| MIND-50K | BERT$_{base}$ | 15.04 | 14.35 | -4.59% |
| HM-50K | ResNet50 | 2.74 | 3.26 | +18.98% |
| | Swin-T | 2.84 | 4.47 | +57.39% |
| Bili-50K | ResNet50 | 1.07 | 1.20 | +12.05% |
| | Swin-T | 1.08 | 1.46 | +35.19% |

## E    MORE RESULTS OF MOREC WITH PRE-TRAINED ME AND ITS TRAINING-FROM-SCRATCH VERSION

We show a more detailed comparison of MoRec with pre-trained ME and its training-from-scratch (TFS) version. We report the convergence behaviors of Pre-trained ME vs TFS on the validation set in Figure 7. It can be found that MoRec with the pre-trained ME have a faster convergence rate and a more stable training process. The performance of TFS is relatively poor and may fluctuate wildly during the convergence process.

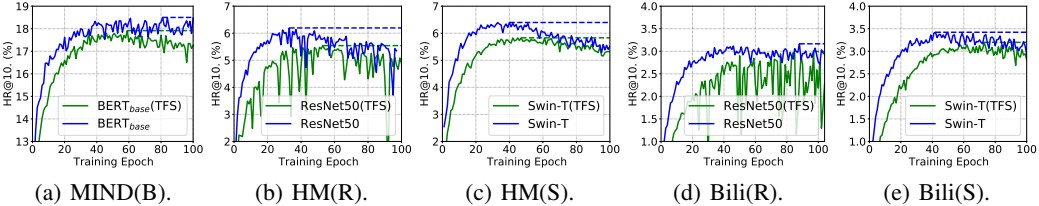

(a) MIND(B). (b) HM(R). (c) HM(S). (d) Bili(R). (e) Bili(S).

Figure 7: Convergence behaviors of pre-trained ME vs TFS on the validation set. B, R, S denotes BERT, ResNet and Swin Transformer, respectively.

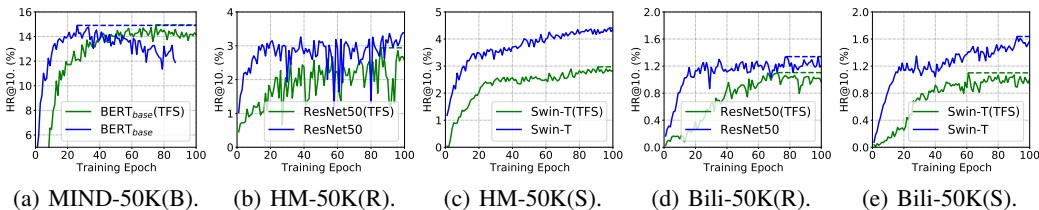

(a) MIND-50K(B). (b) HM-50K(R). (c) HM-50K(S). (d) Bili-50K(R). (e) Bili-50K(S).

Figure 8: Convergence behaviors of pre-trained ME vs TFS on the validation set with a smaller dataset by randomly drawing 50K users. Larger relative improvements can be observed by comparing with the above figure.

We also construct the smaller version datasets by randomly drawing 50K users from MIND, HM, and Bili. We report the recommendation results on the testing set in Table 8 and the convergence behaviors on the validation set in Figure 8. It can be seen that the advantages of pre-trained ME over TFS are more obvious on small datasets. MoRec achieve around 57% and 35% improvements with the pre-trained Swin-T on HM-50K and Bili-50K, respectively. However, we found that the pre-trained BERT$_{base}$ is even worse than its TFS version on MIND-50K.

## F SUPPLEMENTARY EXPERIMENTS

### F.1 MIND DATASET WITH OTHER EVALUATION STRATEGY

The evaluation method we used for MIND differs from the data splitting strategy used in (An et al., 2019; Wu et al., 2021a; Yu et al., 2021). The MIND dataset contains both positive feedback and true negative feedback where items are exposed to the user but there is no observed user interaction. The above works used the exposed but no interacted items as true negative examples rather than the randomly sampled items. However, most recommender system datasets such as HM and Bili contain no item exposure information. For consistency, we treat all these recommender tasks as one-class collaborative filtering (Rendle et al., 2012; Pan et al., 2008)— 99% cases in literature.

Meanwhile, we also conducted experiments in CANO-SC following the same experiment setting of (An et al., 2019; Wu et al., 2021a; Yu et al., 2021), and report the result in Table 9. The result shows that MoRec significantly outperform IDRec in text recommendation as well.

### F.2 INCREASING USER SEQUENCE LENGTH

We conduct additional experiments with sequence length of maximum 23 items in CANO-SC on the HM and Bili datasets, and report the convergence in Figure 10. We find that the results in such settings are consistent as reported before: IDRec and MoRec still show comparable recommendation accuracy.

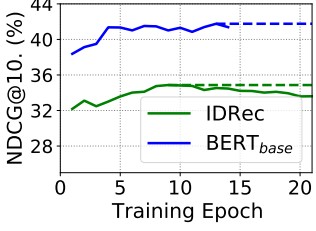
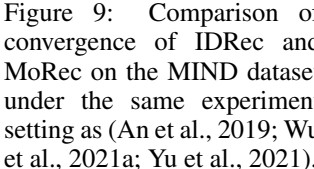
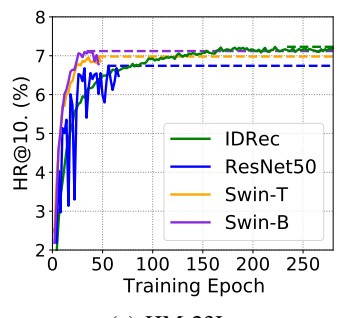
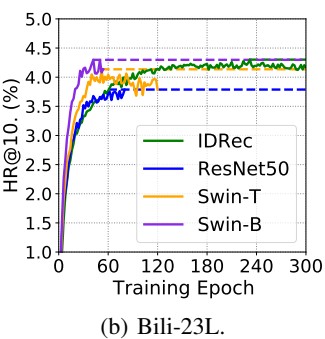

(a) HM-23L.      (b) Bili-23L.

Figure 9: Comparison of convergence of IDRec and MoRec on the MIND dataset under the same experiment setting as (An et al., 2019; Wu et al., 2021a; Yu et al., 2021).

Figure 10: Comparison of convergence of IDRec and MoRec based on the SASRec framework on the validation data of HM and Bili datasets with sequence length of maximum 23 items.

## G  HYPER PARAMETERS AND TRAINING COST

For all methods, we employ an AdamW (Loshchilov & Hutter, 2017) as the default optimizer and find that the dropout rate set to $0.1$ (i.e., removing 10% parameters) offers the optimal results on the validation set. Regarding other hyper-parameters, we follow the common practice and perform extensive searching. For IDRec, we tune the learning rate $\gamma$ from $\{1e\text{-}3, 5e\text{-}4, 1e\text{-}4, 5e\text{-}5\}$, the embedding size $d$ from $\{64, 128, 256, 512, 1024, 2048, 4096\}$, and set batch size $b$ to $1024$ for DSSM and $128$ for SASRec. For MoRec, we set $d$ to $512$ for both DSSM and SASRec, $b$ to $512$ and $64$ for DSSM and SASRec respectively due to GPU memory constraints. Given that ME (e.g., BERT and ResNet) has already well pre-trained parameters, we use relatively smaller $\gamma$ than other parts in the recommender model. That is, we search $\gamma$ from $\{1e\text{-}4, 5e\text{-}5, 1e\text{-}5\}$ for the pre-trained ME networks, and set $\gamma$ to $1e\text{-}4$ for other parts with randomly initialized parameters. Finally, we tune the weight decay $\beta$ from $\{0.1, 0.01, 0\}$ for both IDRec and MoRec. The effectiveness of such factors will be analyzed later in Appendix H.

For the MLPs (multilayer perceptron) used in DSSM, we initially set their middle layer size to $d$ as well and search the layer number $l$ from $\{0, 1, 3, 5\}$ but find that $l = 0$ (i.e., no hidden layers) always produces the best results. For the Transformer block used in SASRec, we set $l$ to $2$ and the head number of the multi-head attention to $2$ for the optimal results. All other hyper-parameters are kept the same for IDRec and MoRec unless specified otherwise. We report details of the pre-trained ME used in the experiments in Table 9 and all hyper-parameter details in the Table 12.

Table 9: Network architecture, parameter size, and download URL of the pre-trained ME we used. L: number of Transformer blocks, H: number of multi-head attention, C: channel number of the hidden layers in the first stage, B: number of layers in each block.

| Modality | Pre-trained model | Architecture | #Param. | URL |
|---|---|---|---|---|
| Text | BERT$_{tiny}$ | L=2, H=128 | 4M | https://huggingface.co/prajjwal1/bert-tiny |
| | BERT$_{small}$ | L=4, H=512 | 29M | https://huggingface.co/prajjwal1/bert-small |
| | BERT$_{medium}$ | L=8, H=512 | 41M | https://huggingface.co/prajjwal1/bert-medium |
| | BERT$_{base}$ | L=12, H=768 | 109M | https://huggingface.co/bert-base-uncased |
| | RoBERTa$_{base}$ | L=12, H=768 | 125M | https://huggingface.co/roberta-base |
| | OPT$_{125M}$ | L=12, H=768 | 125M | https://huggingface.co/facebook/opt-125M |
| Image | ResNet18 | C = 64, B={2, 2, 2, 2} | 12M | https://download.pytorch.org/models/resnet18-5c106cde.pth |
| | ResNet34 | C = 64, B={3, 4, 6, 3} | 22M | https://download.pytorch.org/models/resnet34-333f7ec4.pth |
| | ResNet50 | C = 64, B={3, 4, 6, 3} | 26M | https://download.pytorch.org/models/resnet50-19c8e357.pth |
| | Swin-T | C = 96, B={2, 2, 6, 2} | 28M | https://huggingface.co/microsoft/swin-tiny-patch4-window7-224 |
| | Swin-B | C = 128, B={2, 2, 18, 2} | 88M | https://huggingface.co/microsoft/swin-base-patch4-window7-224 |
| | MAE$_{base}$ | L=12, H=768 | 86M | https://huggingface.co/facebook/vit-mae-base |

Table 10: Searching learning rate $\gamma^M$, $M$ and $R$ denote ME and the remaining modules of SAS-Rec, respectively.

| Dataset | Encoder | $\gamma^R$=1e-4, searching $\gamma^M$ | | |
|---|---|---|---|---|
| | | $\gamma^M$=1e-5 | $\gamma^M$=5e-5 | $\gamma^M$=1e-4 |
| MIND | BERT$_{base}$ | 18.16 | **18.23** | 16.35 |
| HM | ResNet50 | 6.08 | 6.62 | **6.67** |
| | Swin-T | 6.41 | **6.97** | 6.92 |
| | MAE$_{base}$ | 6.26 | **7.03** | 6.96 |
| Bili | ResNet50 | 2.33 | 2.84 | **2.93** |
| | Swin-T | 2.29 | **3.18** | 3.01 |
| | MAE$_{base}$ | 2.78 | 3.05 | **3.18** |

Table 11: Searching same and different weight decay $\beta$ combinations.

| Dataset | Encoder | Searching $\beta^R$ and $\beta^M$ | | | |
|---|---|---|---|---|---|
| | | $\beta^R = \beta^M = \beta$ | | | $\beta^R$=0.1, |
| | | $\beta$=0 | $\beta$=0.1 | $\beta$=0.01 | $\beta^M$=0 |
| MIND | BERT$_{base}$ | 17.31 | 17.96 | **18.23** | 18.19 |
| HM | ResNet50 | 6.42 | 6.36 | **6.67** | 6.57 |
| | Swin-T | 6.51 | 5.88 | 6.81 | **6.97** |
| | MAE$_{base}$ | 6.30 | 5.72 | 6.63 | **7.03** |
| Bili | ResNet50 | 2.76 | 2.58 | **2.93** | 2.74 |
| | Swin-T | 2.73 | 2.47 | 2.76 | **3.18** |
| | MAE$_{base}$ | 2.81 | 2.69 | 2.95 | **3.18** |

Table 12: The best hyper parameters and training cost. $\gamma^R$: learning rate of recommendation network, $\gamma^M$: fine-tune learning rate of modality encoder, $\beta^R$: weight decay of recommendation network, $\beta^M$: weight decay of modality encoder, $d$: embedding size, $b$: batch size, HR: HR@10 (%), ND: NDCG@10 (%), #Param: number of tunable parameters, FLOPs: computational complexity (we measure FLOPs for each model with $b = 1$), Time/E: averaged training time for one epoch, "m" means minutes, BE: epoch with the best validation result, MU: GPU memory use, e.g., "V100-32G(2)" means that we used 2 V100s with 32G memory. BERT$_{base}$(2/12) means we only fine-tune the top 2 blocks out of the 12 blocks, similar to ResNet and Swin Transformer. Note that FLOPs can vary significantly depending on the implementation. For example, we can cache the representations in certain layers of the ME for all items during pre-processing, and avoid the forward&backward propagation on the bottom layers when fine-tuning only the top layers of ME.

| Dataset | Method | $\gamma^R$ | $\gamma^M$ | $\beta^R$ | $\beta^M$ | $d$ | $b$ | HR | ND | #Param. | FLOPs | Time/E | BE | MU | GPU |
|---|---|---|---|---|---|---|---|---|---|---|---|---|---|---|---|
| MIND | IDRec | 1e-4 | – | 0.1 | – | 512 | 128 | 17.71 | 9.52 | 47M | 0.12G | 7m | 120 | 3G | V100-32G(1) |
| | BERT$_{tiny}$ | 1e-4 | 5e-5 | 0.1 | 0 | 512 | 64 | 17.64 | 9.42 | 11M | 0.63G | 10m | 105 | 4G | V100-32G(1) |
| | BERT$_{small}$ | 1e-4 | 1e-4 | 0.1 | 0 | 512 | 64 | 18.50 | 9.94 | 35M | 16G | 42m | 84 | 13G | V100-32G(1) |
| | BERT$_{medium}$ | 1e-4 | 5e-5 | 0.1 | 0 | 512 | 64 | 18.39 | 9.88 | 48M | 32G | 83m | 66 | 23G | V100-32G(1) |
| | BERT$_{base}$(0/12) | 1e-4 | – | 0.01 | – | 512 | 64 | 13.93 | 7.55 | 7M | 0.14G | 3m | 40 | 4G | V100-32G(1) |
| | BERT$_{base}$(0/12)-MLM | 1e-4 | – | 0.1 | – | 512 | 64 | 14.68 | 8.00 | 7M | 0.14G | 3m | 44 | 4G | V100-32G(1) |
| | BERT$_{base}$(2/12) | 1e-4 | 1e-4 | 0.1 | 0 | 512 | 64 | 17.87 | 9.66 | 21M | 107G | 96m | 84 | 27G | V100-32G(1) |
| | BERT$_{base}$(6/12) | 1e-4 | 1e-4 | 0.1 | 0 | 512 | 64 | 18.51 | 10.02 | 49M | 107G | 138m | 89 | 28G | V100-32G(1) |
| | BERT$_{base}$ | 1e-4 | 5e-5 | 0.01 | 0.01 | 512 | 64 | 18.23 | 9.73 | 116M | 107G | 102m | 75 | 52G | V100-32G(2) |
| | BERT$_{base}$-MLM | 1e-4 | 5e-5 | 0.1 | 0 | 512 | 64 | 18.63 | 10.05 | 116M | 107G | 102m | 81 | 52G | V100-32G(2) |
| | RoBERTa$_{base}$ | 1e-4 | 5e-5 | 0.1 | 0 | 512 | 64 | 18.68 | 10.02 | 131M | 107G | 103m | 98 | 53G | V100-32G(2) |
| | OPT$_{125M}$ | 1e-4 | 1e-4 | 0.1 | 0 | 512 | 64 | 17.24 | 9.20 | 132M | 107G | 100m | 39 | 45G | V100-32G(2) |
| HM | IDRec | 5e-5 | – | 0.1 | – | 2048 | 128 | 6.84 | 4.01 | 114M | 1G | 1m | 82 | 5G | V100-32G(1) |
| | ResNet18 | 1e-4 | 1e-4 | 0.01 | 0.01 | 512 | 64 | 6.30 | 3.36 | 18M | 40G | 95m | 28 | 23G | V100-32G(1) |
| | ResNet34 | 1e-4 | 1e-4 | 0.01 | 0.01 | 512 | 64 | 6.40 | 3.40 | 29M | 81G | 136m | 29 | 30G | V100-32G(1) |
| | ResNet50(0/4) | 1e-4 | – | 0.1 | – | 512 | 64 | 4.03 | 2.12 | 7M | 0.09G | 1m | 80 | 4G | V100-32G(1) |
| | ResNet50(1/4) | 1e-4 | 1e-4 | 0.01 | 0.01 | 512 | 64 | 6.46 | 3.45 | 22M | 91G | 110m | 29 | 13G | V100-32G(1) |
| | ResNet50(2/4) | 1e-4 | 1e-4 | 0.01 | 0.01 | 512 | 64 | 6.59 | 3.53 | 29M | 91G | 150m | 37 | 23G | V100-32G(1) |
| | ResNet50 | 1e-4 | 1e-4 | 0.01 | 0.01 | 512 | 64 | 6.67 | 3.56 | 31M | 91G | 83m | 35 | 80G | V100-32G(4) |
| | Swin-T(0/4) | 1e-4 | – | 0.1 | – | 512 | 64 | 3.45 | 1.78 | 7M | 0.07G | 1m | 64 | 4G | V100-32G(1) |
| | Swin-T(1/4) | 1e-4 | 1e-4 | 0.1 | 0 | 512 | 64 | 6.18 | 3.36 | 21M | 96G | 109m | 74 | 43G | V100-32G(2) |
| | Swin-T(2/4) | 1e-4 | 1e-4 | 0.1 | 0 | 512 | 64 | 6.80 | 3.70 | 33M | 96G | 104m | 35 | 54G | A100-32G(2) |
| | Swin-T | 1e-4 | 5e-5 | 0.1 | 0 | 512 | 64 | 6.97 | 3.80 | 34M | 96G | 107m | 35 | 157G | A100-40G(4) |
| | Swin-B | 1e-4 | 1e-4 | 0.1 | 0 | 512 | 64 | 7.24 | 3.98 | 94M | 333G | 102m | 33 | 308G | A100-40G(8) |
| | MAE$_{base}$(0/12) | 1e-4 | – | 0.01 | – | 512 | 64 | 2.50 | 1.26 | 7M | 0.07G | 2m | 68 | 4G | V100-32G(1) |
| | MAE$_{base}$(0/12)-MLM | 1e-4 | – | 0.1 | – | 512 | 64 | 2.79 | 1.40 | 7M | 0.07G | 2m | 73 | 4G | V100-32G(1) |
| | MAE$_{base}$ | 1e-4 | 1e-4 | 0.1 | 0 | 512 | 64 | 7.03 | 3.83 | 92M | 96G | 86m | 42 | 46G | V100-32G(2) |
| | MAE$_{base}$-MLM | 1e-4 | 1e-4 | 0.1 | 0 | 512 | 64 | 7.07 | 3.87 | 92M | 96G | 86m | 42 | 46G | V100-32G(2) |
| Bili | IDRec | 5e-5 | – | 0.1 | – | 1024 | 128 | 3.03 | 1.63 | 72M | 0.25G | 4m | 191 | 4G | V100-32G(1) |
| | ResNet18 | 1e-4 | 1e-4 | 0.01 | 0.01 | 512 | 64 | 2.50 | 1.24 | 18M | 40G | 78m | 37 | 23G | V100-32G(1) |
| | ResNet34 | 1e-4 | 1e-4 | 0.01 | 0.01 | 512 | 64 | 2.73 | 1.37 | 29M | 81G | 113m | 44 | 30G | V100-32G(1) |
| | ResNet50(0/4) | 1e-4 | – | 0 | – | 512 | 64 | 0.72 | 0.34 | 7M | 0.09G | 4m | 51 | 4G | V100-32G(1) |
| | ResNet50(1/4) | 1e-4 | 1e-4 | 0.01 | 0.01 | 512 | 64 | 2.83 | 1.41 | 22M | 91G | 89m | 34 | 13G | V100-32G(1) |
| | ResNet50(2/4) | 1e-4 | 1e-4 | 0.01 | 0.01 | 512 | 64 | 2.89 | 1.43 | 29M | 91G | 116m | 40 | 23G | V100-32G(1) |
| | ResNet50 | 1e-4 | 1e-4 | 0.01 | 0.01 | 512 | 64 | 2.93 | 1.45 | 31M | 91G | 67m | 88 | 80G | V100-32G(4) |
| | Swin-T(0/4) | 1e-4 | – | 0.1 | – | 512 | 64 | 0.79 | 0.37 | 7M | 0.07G | 3m | 55 | 4G | V100-32G(1) |
| | Swin-T(1/4) | 1e-4 | 1e-4 | 0.1 | 0 | 512 | 64 | 2.88 | 1.45 | 21M | 96G | 88m | 53 | 43G | V100-32G(2) |
| | Swin-T(2/4) | 1e-4 | 1e-4 | 0.1 | 0 | 512 | 64 | 3.00 | 1.50 | 33M | 96G | 81m | 47 | 54G | A100-32G(2) |
| | Swin-T | 1e-4 | 5e-5 | 0.1 | 0 | 512 | 64 | 3.18 | 1.59 | 34M | 96G | 86m | 74 | 157G | A100-40G(4) |
| | Swin-B | 1e-4 | 1e-4 | 0.1 | 0 | 512 | 64 | 3.28 | 1.66 | 94M | 333G | 82m | 34 | 308G | A100-40G(8) |
| | MAE$_{base}$(0/12) | 1e-4 | – | 0.01 | – | 512 | 64 | 0.57 | 0.27 | 7M | 0.07G | 2m | 73 | 4G | V100-32G(1) |
| | MAE$_{base}$(0/12)-MLM | 1e-4 | – | 0.1 | – | 512 | 64 | 0.56 | 0.26 | 7M | 0.07G | 2m | 57 | 4G | V100-32G(1) |
| | MAE$_{base}$ | 1e-4 | 1e-4 | 0.1 | 0 | 512 | 64 | 3.18 | 1.60 | 92M | 96G | 78m | 45 | 46G | V100-32G(2) |
| | MAE$_{base}$-MLM | 1e-4 | 1e-4 | 0.1 | 0 | 512 | 64 | 3.17 | 1.61 | 92M | 96G | 78m | 53 | 46G | V100-32G(2) |

## H  LEARNING RATE AND WEIGHT DECAY SEARCHING

**Separate hyperparameters for item and user encoders.** MoRec typically consist of two modules: item encoders or a user encoder. Since item encoders or ME has been already well pre-trained in advance, it might be reasonable to set separate hyper-parameters for ME and user encoder — e.g.. setting a smaller learning rate for ME. We report such results in Table 10 and Table 11. As shown, MoRec in general yield better results by using different learning rates and weight decay for ME and user encoder. In particular, it performs much worse on MIND when applying the same learning rate for all modules (e.g., 16.35 vs. 18.23). Similar behaviors are found for Swin Transformer on the two visual datasets. Thereby, we draw the conclusion that **searching hyperparameters (e.g., learning rate & weight decay) separately for ME & user encoder matters for the best MoRec results.**

## I  TS-BASED MoREC BY ADDING MORE MLP LAYERS

Suggested by reviewer jqDE, we apply several multi-layer perceptrons (MLPs) layers after the dense layer of the fixed representation extracted from the TS-based MoRec. We report the results in Table 13. As shown, we find that deep neural network can indeed improve the performance of TS; however, it is still obviously worse than IDRec and E2E-based MoRec.

Table 13: HR@10 (%) of E2E vs TS with additional MLP layers . "TS-DNN 6" denotes that TS-based MoRec with six additional MLPs layers. 'Improv.(ID)' indicates the relative improvement over IDRec. 'Improv.(TS)' indicates the relative improvement over TS-based MoRec without MLPs.

| Dataset | IDRec | Encoder | TS | TS-DNN | | | | | Improv. (ID) | Improv. (TS) | E2E | Improv. (ID) | Improv. (TS) |
|---------|-------|---------|-----|------|------|------|------|------|--------|--------|------|--------|--------|
| | | | | 2 | 6 | 8 | 10 | 12 | | | | | |
| MIND | 17.71 | BERT$_{base}$ | 13.93 | 15.20 | 16.26 | 16.66 | 16.32 | 16.14 | -5.93% | +19.60% | **18.23** | +2.94% | +30.87% |
| HM | 6.84 | ResNet50 | 4.03 | 4.64 | 5.40 | 5.39 | 5.40 | 5.02 | -21.05% | +33.66% | 6.67 | -2.49% | +65.10% |
| | | Swin-T | 3.45 | 4.46 | 5.28 | 5.55 | 5.40 | 5.38 | -18.86% | +68.87% | **6.97** | +1.90% | +102.2% |
| Bili | 3.03 | ResNet50 | 0.72 | 1.23 | 1.62 | 1.47 | 1.28 | 1.24 | -46.53% | +125.0% | 2.93 | -3.30% | +306.9% |
| | | Swin-T | 0.79 | 1.40 | 1.81 | 2.10 | 1.95 | 1.64 | -30.69% | +165.8% | **3.18** | +4.95% | +302.5% |

## J  CROSS-DOMAIN RECOMMENDATION EVALUATION OF MoREC

Suggested by reviewer jqDE & tNVZ, to evaluate the transfer learning ability of MoRec, we train MoRec on the three large-scale datasets used in this paper as the source domain and then perform fine-tuning on three additional smaller datasets, i.e. the target datasets. To be specific, we use Adressa (Gulla et al., 2017), a Norwegian news recommendation dataset[11] as the target dataset for textual MoRec pre-trained on MIND; Then, we use the Amazon clothing&shoes (Ni et al., 2019) dataset as the target dataset for visual MoRec pre-trained on HM. At last, we use a Kuaishou[12] dataset (collected similarly as Bili) as the target dataset for visual MoRec pre-trained on Bili. For all these target datasets, we randomly sample 20,000 and 5,000 users for evaluation. Note that there is no overlapped users and items between the target and source datasets. The statistics of the target datasets is shown in Table 14.

Table 14: Characteristics of the target dataset for cross-domain recommendations.

| Dataset | $n$ | $m$ | $\mathcal{R}|^{train}$ |
|---------|-----|-----|---------|
| Adressa | $20K$ | 3,149 | $241K$ |
| | $5K$ | 1,633 | $60K$ |
| Amazon | $20K$ | 14,348 | $129K$ |
| | $5K$ | 7,453 | $31K$ |
| Kuaishou | $20K$ | 30,463 | $144K$ |
| | $5K$ | 10,544 | $34K$ |

We report the convergence behavior of MoRec adapted on the target datasets with (w/) and without (w/o) pre-training in Figure 11. First, we indeed can see with pre-training MoRec converges faster on all six datasets, in particular in the beginning epochs; It seems that MoRec with pre-training has more advantage on a smaller target dataset (e.g. on Adressa 5K and Amazon 5K) than on a larger target dataset (i.e. on Adressa 20K and Amazon 20K). Second, pre-trained MoRec does not show obviously better results than its training-from-scratch version except on Adressa, which is a

---

[11]We translate this dataset from Norwegian to English by Google Translate.

[12]https://www.kuaishou.com/?isHome=1

text recommendation dataset. The results surprise us a bit since we thought MoRec could perform substantially better with pre-training in advance. We guess there are two possible reasons: (1) our source dataset is still not large enough compared with the pre-trained data in NLP and CV (BERT and Swin Transformer are trained with huge compute and data sources). It is interesting to see whether MoRec could obtain larger improvements if we scale up the source data from 10x to 100x times larger. (2) The source and target datasets still have some gap (at least for image recommendation) although they have the same modality. We believe this is a very interesting question for future research work. We will release all datasets and codes used for this experiment.

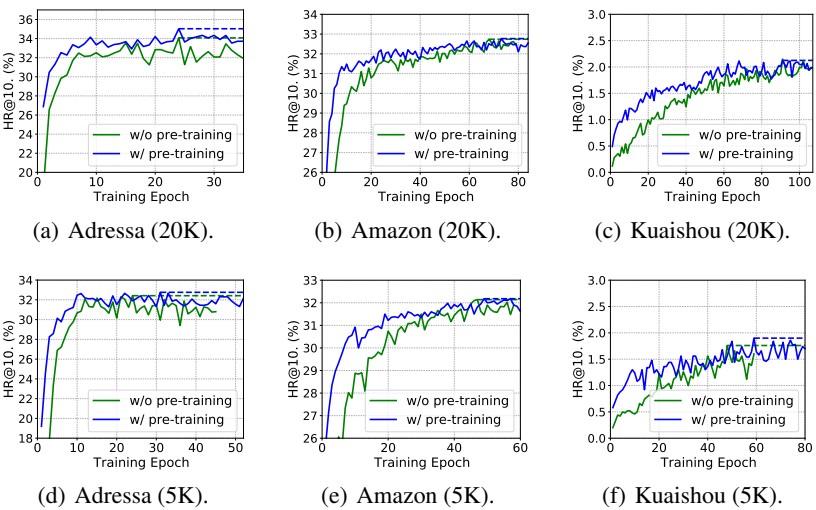

(a) Adressa (20K).     (b) Amazon (20K).     (c) Kuaishou (20K).

(d) Adressa (5K).     (e) Amazon (5K).     (f) Kuaishou (5K).

Figure 11: Comparison of convergence behavior of MoRec w/ and w/o pre-training.

## K   MOREC LITERATURE IN RECENT YEARS

We report the modality-only recommender systems (MoRec) literature in recent years in Table 15 (Note that, in this paper, we study the fair comparison of MoRec (use only modality features) vs IDRec, rather than ID + modality vs IDRec (which is not our focus)). As shown, so far there is no even one MoRec paper performing an explicit study between MoRec and its IDRec counterpart. Note that we do not aim to question these literature since the comparison study might not be a focus for these literature. Corrspondingly, we belive the comparison study of our paper is novel.

Table 15: The modality-only recommender systems (MoRec) literature in recent years.

| Research | Feature | End2end | Fair comparison with IDRec? | Reason |
|---|---|---|---|---|
| HASC (Wu et al., 2019b) | vision | ✘ | ✘ | MoRec & IDRec used different network backbone for comparison |
| NRMS (Wu et al., 2019a) | text | ✘ | ✘ | No comparison with IDRec |
| LSTUR (An et al., 2019) | text | ✘ | ✘ | No comparison with IDRec |
| SAERS (Hou et al., 2019) | vision | ✘ | ✘ | MoRec & IDRec used different network backbone for comparison |
| FIM (Wang et al., 2020) | text | ✘ | ✘ | No comparison with IDRec |
| Prob-BERT (Penha & Hauff, 2020) | text | ✘ | ✘ | No comparison with IDRec |
| KIM (Qi et al., 2021) | text | ✘ | ✘ | No comparison with IDRec |
| SEMI (Lei et al., 2021) | text, vision, video | ✘ | ✘ | MoRec & IDRec used different network backbone for comparison |
| MM-Rec (Wu et al., 2021b) | text, vision | ✘ | ✘ | No comparison with IDRec |
| MINDSim (Luo et al., 2022) | text | ✘ | ✘ | No comparison with IDRec |
| TopicVAE (Guo et al., 2022) | text | ✘ | ✘ | MoRec & IDRec used different network backbone for comparison |
| ACMLM (Ni et al., 2019) | text | ✔ | ✘ | No comparison with IDRec |
| PLM4NewsRec (Wu et al., 2021a) | text | ✔ | ✘ | No comparison with IDRec |
| NewsBERT (Wu et al., 2021c) | text | ✔ | ✘ | No comparison with IDRec |
| UNBERT (Zhang et al., 2021) | text | ✔ | ✘ | No comparison with IDRec |
| MTRec (Bi et al., 2022) | text | ✔ | ✘ | No comparison with IDRec |
| SpeedyFeed (Xiao et al., 2022) | text | ✔ | ✘ | No comparison with IDRec |
| MINER (Li et al., 2022) | text | ✔ | ✘ | MoRec & IDRec used different network backbone for comparison |
| Ours | text, vision | ✔ | ✔ | – |

