# OpenReview forum: "Where to Go Next for Recommender Systems? ID- vs. Modality-based recommender models revisited"
_ICLR.cc/2023/Conference — Submitted to ICLR 2023_

### Official Review · Reviewer_BBQN · 2022-10-27

**Confidence:** 5
**Correctness:** 2
**Technical Novelty And Significance:** 1
**Empirical Novelty And Significance:** 2
**Recommendation:** 3

**Clarity, Quality, Novelty And Reproducibility:**

writing is clear; contributions are not novel; the reproducibility seems easy to implement, though the code is not given.

**Strength And Weaknesses:**

Strong & Weak:

S1: It is interesting to see what happened when the item embeddings are replaced by pre-trained item encoders like BERT and ResNet.

W1: The ModalityRec method proposed in this paper is actually equivalent to the IDRec variants by replacing the item embeddings with pre-trained item encoders like BERT. However, using BERT/ResNet as the item encoder to represent item content has been proposed in Ref [1][2][3]. As a result, the two components of ModalityRec, IDRec architecture + pre-trained item encoders, are both not new. This weakens the contributions of ModalityRec.

References:

[1] Zhang et al, UNBERT: User-News Matching BERT for News Recommendation, IJCAI’21.

[2] Li et al, MINER: Multi-Interest Matching Network for News Recommendation, ACL’22 Findings.

[3] Wu et al, MM-Rec: Multimodal News Recommendation, SIGIR’22


**Summary Of The Paper:**

Summary:

This paper proposes to compare Content-based RecSys, or called Modality-based RecSys in this paper, with ID-based RecSys. The methodology is to replace the item embeddings in IDRec with the pre-trained item encoder in ModalityRec where the item encoder is BERT for item texts and ResNet/Swin-T for item images. Note, the user embeddings are the same for both IDRec and ModalityRec. The paper investigates two neural architectures, one is tower-based DSSM architecture and the other is sequence-based SASRec. The experiments are conducted under three scenarios, i.e., canonical scenario, cold-item scenario, and new-item scenario.


**Summary Of The Review:**

Comments:

C1: Many choices are not given a reason, for example, the numbers 23 and 13 in the following statement, “For MIND, we select the latest 23 items for each user to construct the interaction sequence. For HM and Bili, we choose the 13 most recent interactions...”

---

> ### Author Response · Authors · 2022-11-05
> **Reply2: Dear reviewer BBQN**
>
> Regarding no technical design:
>
> We believe a good paper is not always introducing a novel methodology. In fact, there are several hundreds of such papers in the RS literature. We believe a good paper should help the community to move forward, telling researchers and practitioners unknown but very important facts.  For RS researchers, we believe so far no one knows whether  the SOTA modality-only item recommendation models (MoRec) can beat IDRec under a fair comparison setting (on the non-cold setting)
>
> As we know, there are various types of research work on previous ICLR proceedings, such as a benchmark study, an empirical study, a dataset releasing study or a rethinking study of previous methods. Our paper clearly falls into the empirical study category, and we think it is not always necessary to introduce a new model for such types of papers.  Even some famous papers like ImageNet and [1,2] have not shown technical novelty. **We believe rejecting a paper simply based on no technical design is unfair!**
>
> But **we believe what we are doing in this paper has never been explicitly discussed in the literature (see the table at the top of this page)** and they are very important research questions given the dominant role of IDRec and the recent breakthrough in NLP and CV.  **Note that many comparison papers for IDRec and MoRec are for the cold-start setting, and almost no peer-reviewed papers did an explicit discussion and fair comparison for IDRec vs MoRec.**
>
> In addition, you might find our study is very simple, in fact, it is super expensive. What we are doing in this paper requires a very huge computation（as shown in our appendix) and has taken us up to  **100,000$  for renting GPU servers**. The computational cost and training time of SASRec with Swin-transformer and ResNet is several hundreds of times higher than IDRec (**MoRec requires around **800x longer time than IDRec** using the same GPU hardware when using SASRec +Swin Transformer**).  This work has taken us over 15 months to complete these related experiments.
>
>
>
>
>
>
>
> $\textbf{Q2: Choices are not given a reason: “For MIND, we select the latest 23 items for each user to construct the interaction sequence. }$ $\textbf{For HM and Bili, we choose the 13 most recent interactions...”Why}$
>
> Dear reviewer, we have indeed given the reason just in this sentence: “we choose the 13 most recent interactions **since encoding images requires much larger GPU memory (especially with the SASRec architecture)** ”.   Please also see our paper. Training MoRec by end-to-end learning using SASRec requires 8 most powerful GPUs (A100 40G memory) and can only use a very small batch size. If we use longer sequences, the training is going to be very slow due to a smaller batch size. It requires training for several weeks even for one experiment.
> Despite that we have still added additional experiments  with sequence length 23 for images in our appendix (see Table 10).
>
> $\textbf{Q3:The paper investigates two neural architectures, one is tower-based DSSM architecture and the other is sequence-based SASRec.}$
>
> Although this is not a question,  yet we want to say that DSSM and SASRec can be regarded as two most popular training fashions in current RS literature. There are also some other popular fashions, such as multi-task MMOE, GCN, and various CTR models, but as we have mentioned in our paper, **one paper cannot solve all problems, and the purpose of the research is to inspire more research to do more comprehensive studies.**  It is a start for Modality-only based item recommendation!
>
> [1]He, Rethinking ImageNet Pre-training, CVPR2018
>
> [2] Do Better ImageNet Models Transfer Better? CVPR2019

---

> ### Author Response · Authors · 2022-11-13
> **Reply3: To reviewer BBQN.**
>
> Dear reviewer,
>
> we appreciate your comments. But we find the key point made in the comments is a clear misunderstanding.
>
> You mentioned **using BERT/ResNet as the item encoder to represent item content has been proposed in Ref [1][2][3]. As a result, the two components of ModalityRec, IDRec architecture + pre-trained item encoders, are both not new. This weakens the contributions of ModalityRec.**
>
> Dear reviewer, we never claimed this as a contribution. We did this empirical study just because what you said above is a common practice and then it should be carefully studied.
>
> Second, the three literature you mentioned did not do what we have done in this paper. For example, we focus on IDRec vs.MoRec. But all three literature did not make a careful comparison with a proper IDRec. As we replied before, they used a weak backbone for IDRec but used some advanced modules for MoRec. **There are dozens of such papers in this field and this could easily mislead the community.** Given this, we decided to do a comprehensive and careful empirical study to clarify some factual errors or unfair comparisons in literature. We believe our reply can well explain your question
> Hope you are not an author of the three paper.

---

> > ### Author Response · Authors · 2022-11-16
> > **To reviewer BBQN, about novelty.**
> >
> > Hi all, thanks a lot!
> >
> > Here,**we want to further clarify our novelty and contributions in this paper.**
> >
> > First, we hope clarify a misunderstanding --- we never claimed MoRec iteself  was  a contribution by replacing ID embedding with modality encoder (ResNet, BERT)**. This is a common way and that is the reason we want to carefully study it. We guess reviewer BBQN might have a misunderstanding.
> >
> > Second, we want to say, although there are some  peer-reviewed literature that proposed new models or claimed new SOTA. But unfortunately, **none of them has done a fair comparison, i.e. comparing their proposed MoRec to a proper IDRec**($\textcolor{red}{a \ fundamental \ question}$).  In fact, much literature used  a weak backbone for IDRec as baseline (we have listed these papers below). Since no published paper has done a fair comparison for MoRec (modality-**only**) and IDRec, we believe our work has novelty. Our findings are very important for guiding new research in the community.  In fact, ICLR2022,2021 and NeurIPS had accepted many such empirical study papers.
> >
> > Finally,we want to clearly show our key findings, we believe they are new!
> >
> > (1) We are the first to find that MoRec with DSSM backbone is not a good choice, falling far short of IDRec's performance.  **No paper has told the community the backbone network should be carefully considered when replacing ID with modality encoder.** We believe this is new and important for researchers of this direction.
> >
> > (2) We are the **first** to empirically show that  MoRec （SASRec +Swin Transformer) can **achieve near SOTA (comparable with IDRec for image recommendation (on the non-cold setting).**  This is a very important finding for modality-only reecommendation. There are actually no peer-reviewed MoRec (modality-only) paper by E2E learning for image RS. In fact,**we could not even find a peer-reviewed paper that $\textcolor{red}{use \  only \ image \  features \  }$ for RS.**
> >
> >
> > (3) We are the first to clearly demonstrate that MoRec (SASRec+ BERT, using only  text features)  can already outperforms IDRec. No one has provided a fair comparison in peer-reviewed literature. We also show that MoRec with ResNet still cannot outperform IDRec.
> >
> > (4) We are the first to benchmark so many modality encoder's for MoRec and found that some advances made in NLP and CV could be inherited by RS community, this is very important findings. It is not easy to perform such research since for most image modality encoders (e.g. ResNet, Swin Transformer), MoRec has to be trained with **100x-1000x** longer time & compute.
> >
> > (5) We gave both positive and negative facts, e.g. by replacing modality encoder with ever-bigger BERT, MoRec cannot be always improved. By performing a second round of pre-training on image item encoder，E2E MoRec cannot be improved. Without experiments, we cannot know them.
> >
> >
> > (6) Some researchers may know that E2E usually performs better than the 2-stage training. However, so far there is no even one E2E MoRec  that is successfully deployed in a real system according to literature. In this paper, we highlight that the widely used two-stage method will cause **non-negligible performance drop**. This is what we want to tell the industrial practitioners, in particular for image RS (there is no peer-reviewed E2E paper for MoRec (modality-only) RS so far). We also show that E2E requires nearly 1000x longer training time, 300x larger compute. This is not  explicitly mentioned in previous E2E literature.
> >
> > (7)We present several other findings in section 7, some are widely used but not work, some are unknown such as using different learning rate for user &  item encoder.
> >
> > (8) **We will release a large RS dataset with raw image features which can be used for E2E training.** This dataset contains rich image contents, which is vastly different from HM that only includes a single object i.e., clothes and shoes. Images in our Bili look more difficult to understand compared with HM  (see our appendix, very clear). We believe a large-scale high-quality dataset of great value for this community.   **In fact, we have collected over 2 million users, 150,000 images, and over 50 million interactions ($\textcolor{red}{a \  very \ huge \ dataset}$) for over a year (It has took us a year to collect so much data with a lot of servers). We promise to make this dataset publicly available if this paper is accepted!  We believe even such a large, unique and high-quality dataset can be an ICLR paper like ImageNet (a large number of research work (RS & CV) can be inspired).**
> >
> > **For this paper, it includes empirical study, revisiting study, item encoder benchmark and dataset releasing.** Hope we can convince you. We have spent up to 100,000 dollars and over 15 months doing these empirical study. We believe our paper;'s contribution is even more important than designing a new RS model.**
> >
> >
> >
> >
> >
> > [1] Elsayed et al, End-to-End Image-Based Fashion Recommendation. pre-print 2022

---

> > > ### Author Response · Authors · 2022-11-20
> > > **To reviewer BBQN, we believe we have clearly clarified your question.**
> > >
> > > We want to use a few sentences to highlight why we are new.
> > >
> > > (1) This paper we investigate **modality-only item recommendation rather than $\textcolor{red}{modality + ID}$**.  There are only a few such papers and **not even one image recommendation paper (at least evaluate on a medium size dataset rather than a toy dataset with several hundreds of users) that uses only raw images as features and dares to perform E2E learning or compare to a proper IDRec baseline.**  No peer-reviewed paper has demonstrated that MoRec can achieve comparable results with IDRec by only using modality features (neither text nor images). This is the reason we argue we are new.
> > >
> > > (2) We highlight the fair comparison issues. Although some papers compared with some IDRec, but they did not fairly compare them, i.e. at least make sure the baseline IDRec has the same backbone as their proposed  MoRec, and make sure the training method is fair, rather than using different loss and sampling method for these models. The comparison of MoRec and IDRec is very important. **After MoRec (content-based Rec) was beated by IDRec (collaborative filtering) about ten years ago, no even one paper has shown that MoRec (content-only) can compare with IDRec during the 10 years. This is the value of our paper**
> > >
> > > The two points make our paper very new. Besides, as we mentioned, there are no large-scale suitable image recommendation datasets. Although HM (released in 2022) has raw images, yet they are all about product images (e.g. a pair of shoes or a piece of clothing)  and are relatively easy to be understood by deep learning. These methods cannot be easily generable to other more complex image recommendation scenarios, such as the image dataset we present (see Appendix image cases) in this paper. Note that we will also release a large image recommendation dataset which includes various raw images. We promise to release our final dataset version, which includes **2 million users, 150,000 raw images, and over 50 million interactions**. This will greatly advance this field, even largely advance the CV field (treating RS as an important CV evaluation task, a lot of new research can be done).
> > >
> > > At last, a very short summary of the implication of this paper: **We clearly tell the recommender system community that after $\textcolor{red}{almost \  20 \ years' \ developments \ in \ NLP \  and \ CV \ communities}$,  their most advanced representation encoders' (i.e., BERT and Vision Transformer) expressive ability can finally achieve comparable results with the basic ID embedding in the RS field. $\textcolor{red}{The \ dominant \ role \ of \ IDRec \ could  \ be \  shaken \ in  \ the \ near future}$. No even one peer-reviewed paper has told the community about this. This is a big progress, implying that the key research in the RS community might a chance to shift from IDRec to MoRec.  $\textcolor{red}{Donot \ you \ think \ this \ is \ the  \  most \ important  \  thing \ for \ the \ RS \ community}$? To help this happen more quickly, we would release a huge dataset as mentioned above with 50 million user-image interactions. The large-scale dataset will definitely inspire our community to have a series of pioneering work which will enhance the connection of RS and CV.**

---

### Official Review · Reviewer_j7gn · 2022-10-31

**Confidence:** 4
**Correctness:** 4
**Technical Novelty And Significance:** 2
**Empirical Novelty And Significance:** 3
**Recommendation:** 8

**Clarity, Quality, Novelty And Reproducibility:**

The paper is clear written and easy to follow. The idea is not necessarily quite novel, but it is a meaningful question to revisit at this time.

**Strength And Weaknesses:**

The paper revisited an important question as they observe IDRec model are becoming more prevalent in industry. To compare it with MoRec models, they proposed several questions to examine the two models from diverse perspectives, and performed many experiments, then gave suggestion/ conclusion based on their observations. In section 8, they also provided some additional observations they had, which provided many practical suggestions especially helpful for people who consider leveraging pre-trained models in their MoRec models.

Question:
In the IDRec model that the author used in their experiment, it is not clear to me are the ID embeddings randomly initialized and trained E2E in the big model, or there are some pre-trained embeddings used for the items?


**Summary Of The Paper:**

In this paper, the authors revisited the topic: whether the modality based recommender models (MoRec) can exceed or be on par with the ID-only based models (IDRec) when item modality features are available? They examined this question from several perspective, and performed multiple experiments. Their results show that MoRec with standard end-to-end training is highly competitive and even exceeds IDRec in some cases.

**Summary Of The Review:**

The paper studies an important topic. The authors aim to compare the ID-based recommendation models and modality based recommendation models. They did the comparison in multiple fold and draw the conclusions based on their observations from multiple experiments. The logic is clear and the conclusion is helpful for creating recommendation models in industry.

---

> ### Author Response · Authors · 2022-12-07
> **To reviewer j7gn. Thanks for your great support. We might need more of your help.**
>
> Dear reviewer，
>
> We appreciate your great support!   However,  some reviewers questioned our novelty because they thought there were already some similar papers. We believe they have some misunderstandings!  We assure that  a **real comparison study** between ID  & modality has never been studied in prior literature (**see a statistical Table on top of this page, for all reviewers**).
>
> In fact, we noticed that most prior literature usually used a weak backbone network for IDRec, but used more advanced tricks for their MoRec e.g. the three literature [1,2,3] listed by reviewer BBQN --- the first two use FM and DeepFM for IDRec as a baseline, but their proposed MoRec used attention module, and then they claimed the got SOTA. **This is a very unfair comparison.**  In this case, we cannot draw the conclusion that MoRec achieve SOTA, or MoRec with modality encoder is more powerful than IDRec, right?  **As an AI researcher, we know that there are many factors (network architectures, sampling, loss, etc) that could affect the results.**
>
> Without a fair comparison, the SOTAs do not make much sense, instead will mislead our community. A lot of other researchers (in particular the new researchers) will easily follow such ways and do more problematic papers. In fact,  several recent papers [4,5,6] mentioned that many SOTA in our community is questionable, and the SOTA is still the one from several years ago. When there are too many flawed papers in the peer-reviewed literature, we feel worried about the impact and reputation of our RS community.
>
> **Some reviewers mentioned this paper has no technical designs**,  but we know this is determined by the type of our paper. As we can see, this is a pure empirical study paper, and for such papers, we think the technical design might not always be necessary. By contrast, such papers often focus on other aspects. In fact, we find in ICLR, NIPS, and CVPR, many papers are doing an empirical study, benchmark study, dataset study, or rethinking study. Many of them have no technical novelty like us, e.g. ImageNet and [7,8] (very famous). But we find that most such papers are about NLP or CV, with very few RS papers.  Why should we be so mean to researchers in our own field. Why cannot we be more open? Hope you can help us!
>
> A community should not make all papers focus on a technical design or SOTAs. Without a revisiting or benchmark study, many wrong or even fraudulent papers could mislead the community. This is the reason we explicitly discuss IDRec and MoRec, and we believe **we are the first to find that with an advanced SASRec architecture, MoRec with the most  powerful modality encoder (i.e., Swin Transformer) can finally compete with IDRec, this is unexplored before**
>
>  In fact, as we mentioned (on the top of this page for all reviewers), performing our study is not easy. Many reviewers did not realize **MoRec with E2E training requires $\textcolor{red}{100-1000\times}$ larger training time and compute, many university teams cannot do such expensive experiments.** In addition, many backbone networks (e.g. DSSM and similar CTR models) do not work well by replacing ID embedding with modality encoder. Even using SASRec + Swin Transformer we can obtain only comparable results with IDRec, but such results are not easy to be published on a top-tier venue given no beautiful improvements. These factors largely hinder the progress of MoRec. **With so many barriers, MoRec has made much smaller progress even in recent years compared with  multi-modal learning in NLP and CV.**
>
> As you mentioned, our paper's findings provided many useful findings and practical suggestions for  MoRec research. Would you like to help us persuade other reviewers (**without your support, this paper has no hope**)?
>
> **Q:  it is not clear to me are the ID embeddings randomly initialized and trained E2E in the big model, or there are some pre-trained embeddings used for the items?**
>
> Thanks for your question. All ID embedding used in this paper are trained from scratch ie. randomly initialized. There are some ways to do transfer learning by finetuning a large pre-trained IDRec model to adapt to the new RS scenario. However, IDRec has to rely on user or item ID overlap to perform such transfer learning, a very key drawback of IDRec.
>
> [1] Zhang et al, UNBERT: User-News Matching BERT for News Recommendation, IJCAI’21.
>
> [2] Li et al, MINER: Multi-Interest Matching Network for News Recommendation, ACL’22 Findings.
>
> [3] Wu et al, MM-Rec: Multimodal News Recommendation, SIGIR’22
>
> [4] Rendle et al. On the difficulty of evaluating baselines: A study on recommender systems. Recsys2020
>
> [5] Krichene et al.  On sampled metrics for item recommendation. KDD2020 Best paper
>
> [6]Dacrema et al. Are we really making much progress a worrying analysis of recent neural recommendation approaches. RecSys2019
>
> [7] He et al. Rethinking ImageNet Pre-training, CVPR’2018
>
> [8] Do Better ImageNet Models Transfer Better? CVPR'2019

---

### Official Review · Reviewer_eDRW · 2022-11-01

**Confidence:** 5
**Correctness:** 3
**Technical Novelty And Significance:** 2
**Empirical Novelty And Significance:** 2
**Recommendation:** 5

**Clarity, Quality, Novelty And Reproducibility:**

The technical contribution of this paper is limited. This paper is more concerned with proposing a general framework than with technical contribution. Overall speaking, the paper is well presented.

**Details Of Ethics Concerns:**

There is no ethic concern.

**Strength And Weaknesses:**

Strengths.
1. This paper is about recommender systems, which is one of the most successful applications of deep learning, and thus this paper well matches the scope of ICLR.
2. The authors have put a lot of effort into collecting two large-scale datasets, which consist of both adequate behavioral logs and modality data. The public release of these two datasets, as promised by the authors, will benefit the community.
3. The paper writing is good. The figures and tables are clearly presented and easy to understand.

Weaknesses
1. The technical contribution is limited. For an ICLR paper, more than just proposing a general framework is required (it is quite easy for recommendation engineers or researchers to propose the framework if they are required to utilize modality in recommendation).
2. The experimental results are not fully explained and analyzed. For example, the performance drop in DSSM when modality is utilized is not well explained. I prefer to attribute it to the weakness of the DSSM model rather than the non-sequential setting.
3. Some experiments are not helpful since the conclusions are apparent. For example, it is obvious that end-to-end training will have better performance than two-stage training.

**Summary Of The Paper:**

This paper on recommender systems approaches the question of the utility of modality in recommendation. The authors carefully collected two large-scale datasets of multimedia. Experiments on two collected datasets and one public dataset verify that leveraging modality can benefit the recommendation performance on both sequential and non-sequential recommendation. The authors also provide some insights based on the comparisons among different feature-extraction methods (neural network models for textual input and vision input).

**Summary Of The Review:**

The paper proposes a general framework for leveraging modality for recommendation, but the technical contribution is limited.

---

> ### Author Response · Authors · 2022-11-11
> **Reply3: To reviewer eDRW. Regarding novelty.**
>
> **Q4: Some experiments are not helpful since the conclusions are apparent. For example, it is obvious that end-to-end training will have better performance than two-stage training.**
>
> Thanks.  We summarize some key findings as below.  Hope we can address your concern.  Please kindly note that this paper focuses on **modality-only RS rather than modality+ID RS (this type of research is not new)**.
>
> Imagining that:
>
> **(1)** If we did not clearly demonstrate that MoRec with DSSM (by replacing modality encoder with ID embeddings) shows very poor results. Would other researchers in our community know about this?   Many of them might try their MoRec with DSSM or other CTR architecture but find it did not work at all and then stop such research.  Our paper clearly told the community that the user encoder network matters a lot for MoRec.  We believe this empirical finding is new and important to guide future research.
>
> **(2)** If we did not show MoRec **achieve near SOTA (comparable with IDRec **for the first time**) for image recommendation (on the non-cold setting).**, will our community know this? In fact, there is no  peer-reviewed paper did E2E training for image-**only** item recommendation. In fact, we  can hardly find even an $\textcolor{red}{image-only}$  RS paper in literature for the non-cold RS setting.  In view of this, our paper is very new.
>
> **(3)**  Will the RS or CV community knows that even a powerful ResNet can still not outperform a simple ID embedding?  We also benchmarked  several famous modality encoder for the RS task.
>
>
> **(4)**  If we did not explicitly show the cost of E2E learning, will other researchers know it takes **800x longer training time and 300x  compute?** That is the reason so far not even one paper mentioned a successful E2E model was deployed in an industrial system.
>
>
> **(5)** As you mentioned most researcher know that E2E is better. We agree with this. But our point did not focus on telling the readers about this. We want to tell the practitioners, despite the 2-stage strategy is a common practice for industrial systems,  it will cause a **non-ignorable performance drop** (**4x worse than E2E** in Bili). We also demonstrated that its training time and compute are 100x- 1000x larger than IDRec. This has not been clearly discussed in previous  literature (in particular for the image RS). A lot of researchers and practitioners will not realize these issues without explicitly showing them.
>
> **(6)** Without extensive results, do other researchers know for sure that the improvements in NLP and CV can **in general** be translated to improve the performance of RS. We actually showed both positive and negative observations.
>
> **(7)** There are indeed some negative findings, we did not present them given limited space. E.g. (1) do anyone  know that if we concat the representation of BERT/ResNet with an ID embedding (can also add some DNN layer on top), it has almost no any improvements --- still not sure about the reason; (2)  do anyone  know that if we concat both BERT &  ResNet representation by a basic strategy (add one or two DNN layers on top), MoRec cannot be improved. Many intuitive ideas do not work as expected. Without doing these rigorous experiments, no one knows that.
>
>
> **(8)** We also showed the learning rate should be different for the user and item encoder, for some dataset, if you use the same learning rate MoRec could be very bad.
>
> **(9)** **We will release a large image dataset with raw image features that can be used for E2E training.** In fact, we have collected over 2 million users, 150,000 images, and over 50 million interactions (a very huge dataset) for over a year. We promise to make this dataset public if this paper is accepted! We believe even such a large dataset's release can be an ICLR paper.
>
> Finally, we agree the paper has somewhat limitation, e.g. we did not evaluate all famous RS models and modality encoders.  But we believe the purpose of this paper is not to **solve all problems, but tell the community some important facts and inspire more new  research .** As we mentioned,  this paper has taken us over 15 months and 100,000$ to do these experiments. We cannot evaluate all possible scenarios, which needs the efforts of the whole community if we plan to completely replace IDRec. The other limitation as we mentioned in the conclusion section is that we are still unsure  **whether our findings hold or not if we scale up the dataset to 100-1000x larger like real industry datasets in Google or Amazon*.  We believe such empirical study should deserve equal attention to the paper proposing new methods.
>
>
>
>
>
>
>
>
>
> [1] Zhang et al, UNBERT: User-News Matching BERT for News Recommendation, IJCAI’21.
>
> [2] Li  et al, MINER: Multi-Interest Matching Network for News Recommendation, ACL’22 Findings.
>
> [3] Wu  et al, MM-Rec: Multimodal News Recommendation, SIGIR’22
>
> [6 ] Shereen et al. End-to-End Image-Based Fashion Recommendation, preprint

---

> > ### Author Response · Authors · 2022-11-19
> > **Reply3: To reviewer eDRW, a short summary of our novelty**
> >
> > We want to use a few sentences to highlight why we are new.
> >
> > (1) This paper we investigate **modality-only item recommendation rather than $\textcolor{red}{modality + ID}$**.  There are only a few such papers and **not even one image recommendation paper that uses only raw images as features and dares to perform E2E learning or compare to a proper IDRec baseline.**  No peer-reviewed paper has demonstrated that MoRec can achieve comparable results with IDRec by only using modality features (neither text nor images). This is the reason we argue we are new.
> >
> > (2) We highlight the fair comparison issues. Although some papers compared with some IDRec, but they did not fairly compare them, i.e. at least make sure the baseline IDRec has the same backbone as their proposed  MoRec, and make sure the training method is fair, rather than using different loss and sampling method for these models. The comparison of MoRec and IDRec is very important. **After MoRec (content-based Rec) was beated by IDRec (collaborative filtering) about ten years ago, no even one paper has shown that MoRec (content-only) can compare with IDRec during the 10 years. This is the value of our paper**
> >
> > The two points make our paper very new. Besides, as we mentioned, there are no large-scale suitable image recommendation datasets. Although HM (released in 2022) has raw images, yet they are all about product images (e.g. a pair of shoes or a piece of clothing)  and are relatively easy to be understood by deep learning. These methods cannot be easily generable to other more complex image recommendation scenarios, such as the image dataset we present (see Appendix image cases) in this paper. Note that we will also release a large image recommendation dataset which includes various raw images. We promise to release our final dataset version, which includes **2 million users, 150,000 raw images, and over 50 million interactions**. This will greatly advance this field, even largely advance the CV field (treating RS as an important CV evaluation task, a lot of new research can be done). If our reviewers do not accept this paper, I cannot understand why? Is it because there are some conflicts of interest? or Is it because our reviewers' papers here have been questioned?
> >
> > At last, a very short summary of the implication of this paper: **We clearly tell the recommender system community that after $\textcolor{red}{almost \  20 \ years' \ developments \ in \ NLP \  and \ CV \ communities}$,  their most advanced representation encoders' (i.e., BERT and Vision Transformer) expressive ability can finally achieve comparable results with the basic ID embedding in the RS field. $\textcolor{red}{The \ dominant \ role \ of \ IDRec \ could  \ be \  shaken \ in  \ the \ near future}$. No even one peer-reviewed paper has told the community about this. This is a big progress, implying that the key research in the RS community might a chance to shift from IDRec to MoRec.  $\textcolor{red}{Donot \ you \ think \ this \ is \ the  \  most \ important  \  thing \ for \ the \ RS \ community}$? To help this happen more quickly, we would release a huge dataset as mentioned above with 50 million user-image interactions. The large-scale dataset will definitely inspire our community to have a series of pioneering work which will enhance the connection of RS and CV.**

---

### Official Review · Reviewer_tNVZ · 2022-11-02

**Confidence:** 4
**Correctness:** 3
**Technical Novelty And Significance:** 2
**Empirical Novelty And Significance:** 2
**Recommendation:** 6

**Clarity, Quality, Novelty And Reproducibility:**

The paper has good readability and clarity in terms of paper writing and empirical analysis. However, the conclusions drawn from their experiments are not exciting. It would be more interesting to investigate some novel questions, e.g., cross-domain recommendations, and the contribution of modality information to recommendation interpretability.

**Strength And Weaknesses:**

Strongness
1. The paper is well-written and easy to follow. The motivation for investigating the comparison between MoRec and IDRec is intuitive and clear.
2. Extensive experiments draw contributive conclusions and show researchers inspiring findings about MoRec.

Weakness:
1. Experiments based on DSSM and SASRec seem not sufficient since recommendation accuracy highly relies on the model details, e.g., network designs, score prediction by FFN or dot-product computation.
2. Although extensive experiments demonstrate and conclude many useful findings, those conclusions are not surprising and insightful. The authors should provide more analyses and insights from the experimental results.


**Summary Of The Paper:**

The paper raises the research question of whether the modality-only based models (MoRec) can exceed or be par with the ID-only based models (IDRec). To answer the question, the authors study the performance of MoRec and IDRec under different scenarios, e.g., item cold-start and new item scenarios, and under different settings. Accordingly, they conducted rigorous experiments for item recommendations with two modalities based on two recommendation models. The useful conclusion is that MoRec can benefit from the modality representation techniques in NLP and CV and potentially improve recommendation performance.

**Summary Of The Review:**

The paper provides extensive experiments to investigate several research questions about the contribution of the modality content to a recommendation. It contributes to the recommendation community but lacks novelty and originality and insightful conclusions.

---

> ### Author Response · Authors · 2022-11-12
> **Reply2: To reviewer tNVZ.**
>
> **Q2：Experiments based on DSSM and SASRec seem not sufficient since recommendation accuracy highly relies on the model details, e.g., network designs, score prediction by FFN, or dot-product computation.**
>
> Thanks, we agree that there are more recommender backbones, but as we mentioned in our paper (conclusion section), one paper cannot solve all the problems and we do not expect using only a paper to change the dominant techniques in the community, especially given IDRec has dominated the community for over 15 years. But we are inspiring new research, hoping more papers can contribute. We believe some facts revealed in our paper are valuable although we did not test all backbones. As we mentioned, E2E training could take nearly 1000x longer training time and 300x large compute, we cannot evaluate all possible situations. But in general, DSSM and SASRec can be regarded as representative but very different backbones in the current RS community. Other backbones like Graph NN, Multi-task, CTR models like Wide and Deep, NFM, and YoutubeDNN cannot be all evaluated in one paper. But based on our empirical result, other researchers can roughly guess that MoRec with these backbones cannot compare IDRec because their training manner is more like DSSM rather than SASRec ---- which is a sequence-to-sequence training manner (it has multiple loss, one position per loss) and thus much more powerful in doing the next item prediction task.
>
> What you suggested about using different scoring functions ( FFN vs. dot-product) is very interesting.  As we know, recently there is a paper (NCF vs MF [4]) by Steffen Rendle et who did some empirical study and found that FFN (i.e. NCF) cannot outperform dot product. Now, most IDRec papers use the basic dot product scoring function.
> But that paper only investigates IDRec, which score is more powerful if we replace IDRec with MoRec, **will Rendle's paper hold or He's ([5]) paper hold? This is a very interesting question**. We think it is important to investigate --- we are indeed doing this as independent work.  Some initial results show that the FFN could be better than MF  in several cases when using MoRec. But this is kind of beyond our scope in this paper ---  it needs much more experiments to back up the claim.  We may release this work in the beginning of next year. In fact,  there are some findings made for IDRec could not hold well when we replace ID embedding with modality encoder, but this means more research should be done if we want to shift the mainstream paradigm from IDRec to MoRec.
>
> Hope we can convince you and hope you can help us persuade other reviewers.
>
> **The most important finding of this paper is that we are the first to clearly tell the community that Modality-only recommendation model (MoRec) can show comparable results with IDRec in the non-cold setting. Before our paper, no one has explicitly claimed this, and no one had done a fair comparison.**
> We believe our findings will largely inspire new work in E2E MoRec. We also revealed the key problem for MoRec research, such as huge compute and training costs, lack of available datasets, scaling effect, etc.
>
> As for investigating cross-domain RS, we will show the results in our appendix, but we think the cross-domain RS issue can be a separate paper.
>
>
> Reference:
> [1] Zhang et al, UNBERT: User-News Matching BERT for News Recommendation, IJCAI’21.
>
> [2] Li et al, MINER: Multi-Interest Matching Network for News Recommendation, ACL’22 Findings.
>
> [3] Wu et al, MM-Rec: Multimodal News Recommendation, SIGIR’22
>
> [4] Rendle et. Neural Collaborative Filtering vs. Matrix Factorization Revisited. RecSys2020
>
> [5] He et al. Neural Collaborative Filtering. WWW2017

---

> ### Author Response · Authors · 2022-11-13
> **Reply3: To reviewer tNVZ.**
>
> Dear reviewer tNVZ, thanks again
>
> Here,**we want to further clarify our novelty and contributions in this paper.**
>
>
> First, we want to say, although there is some peer-reviewed literature that proposed new models and claimed SOTA. But unfortunately, none of these formally published papers has done a fair comparison, i.e. comparing their proposed MoRec with a proper IDRec. In fact, much literature used a weak backbone for IDRec or use a bad sampler (we listed such papers at the top of this page). Since no published paper has done a fair study for MoRec and IRec, we believe our work is very meaningful considering so many papers do not want to show the real progress of MoRec. Our study and conclusion are very important for guiding new research in the community.  In fact, ICLR NIPS and CVPR had accepted many famous empirical study papers [2,3 ] although they have no technical contribution.
>
> Finally, we want to clearly state our key findings.
>
> (1) We are the first to find that MoRec with DSSM backbone is not a good choice, falling far short of IDRec's performance.  **No paper has told the community the backbone network should be carefully considered when replacing ID with modality encoder.** We believe this is new and important for researchers of this direction.
>
> (2) We are the first to empirically demonstrate that  MoRec （SASRec +Swin Transformer) can **achieve near SOTA (comparable with IDRec for the first time) for image recommendation (on the non-cold setting).**  This is a very big progress. There are no peer-reviewed MoRec (image-only) papers using E2E learning except [1] (still a preprint). Like other literature, [1] did not compare with IDRec and used a 20x smaller dataset than ours.
>
> (3) We are also the first to demonstrate that MoRec with text modality encoder can already outperform IDRec. Although there are a few papers doing E2E text MoRec, no one has clearly demonstrated this.
>
> (4) We are the first to benchmark so many modality encoders for MoRec and found that some improvements made in NLP and CV could be inherited by RS community, this is very important findings. It is not easy to perform such research since for most image modality encoders, MoRec has to be trained with **100x-1000x** longer time and compute. Our findings in so many large datasets are new.
>
>
> (5) We demonstrated both positive and negative facts, e.g. by replacing modality encoder with ever-bigger BERT, MoRec cannot always be improved. By performing a second round of pre-training on image item encoder， E2E MoRec can be improved.
>
>
> (6) Although some researchers know E2E could perform better than 2-stage training. However, so far there is no even one E2E MoRec that is successfully deployed in a real system according to literature. In this paper, we highlight that the widely used two-stage method will cause **non-negligible** performance drop. This is what we want to tell the industrial practitioners, in particular for image RS (there is no peer-reviewed E2E paper for image RS). We also show that E2E requires nearly 1000x longer training time, and several hundreds of times larger compute. This is not clearly mentioned in previous E2E literature.
>
> (7)We present several other tricks in section 7, some are widely used but do not work, and some are unknown such as using different learning rate for user and item encoder.
>
> (8) **We will release a large RS dataset with raw image features that can be used for E2E training.** To our best knowledge, this is the only RS dataset that contains rich image contents, which is vastly different from HM that only includes a single object i.e., clothes and shoes. Images in our Bili look more difficult to understand compared with HM  (see our appendix, very clear). We believe a large-scale high-quality dataset is of great value for this community.   **In fact, we have collected over 2 million users, 150,000 images, and over 50 million interactions ($\textcolor{red}{a \  very \ huge \ dataset}$) for over a year. We promise to make this dataset public if this paper is accepted!  We believe even such a large and unique dataset can be an ICLR paper. (More research work can be inspired)**
>
> **For this paper, it includes empirical study, revisiting study, item encoder benchmark, and dataset releasing.** Hope we can convince you to accept it. We have spent up to 100,000$ dollars and over 15 months doing these empirical studies. We believe our paper;'s contribution could be even more important and valuable than developing a new RS model.**
>
>
>
> [1] Elsayed et al, End-to-End Image-Based Fashion Recommendation. pre-print 2022
>
> [2] He, Rethinking ImageNet Pre-training, CVPR’2018
>
> [3] Do Better ImageNet Models Transfer Better? 2019

---

### Official Review · Reviewer_jqDE · 2022-11-03

**Confidence:** 4
**Correctness:** 4
**Technical Novelty And Significance:** 1
**Empirical Novelty And Significance:** 2
**Recommendation:** 5

**Clarity, Quality, Novelty And Reproducibility:**

Clarity: The paper is very pleasant to read. The motivations are well explained. All the experiments are well structured and the results are clear.

Quality & Reproducibility: Codes are not available yet. The experimental setup seems fair (in term of hyper-parameter search) and the results are reliable.

Novelty: There is no technical novelty in the paper.

**Strength And Weaknesses:**

Strength. The authors propose an empirical comparison of MoRec and IDRec. The experiments are serious, exhaustive and well conducted. In particular, the authors chose to compare:
- 2 model architectures: DSSM (two-tower model) and SASRec (sequential model)
- several encoders including BERT, RoBERTa, ResNet and Swin
- 3 datasets, including one new dataset (Bili) on video recommendation - MIND (text), H&M and Bili (images)
- several learning strategies: end-to-end training, frozen encoders, etc.
This kind of empirical comparison is useful for the recsys community.

Weaknesses. There is no technical novelty in the paper and the empirical novelty is somewhat limited. There is a recent trend in integrating the recent advances of NLP to build a general recommender system. It would be great to include some technical challenges to improve the technical part of the paper, or to quantify the transfer skills of the MoRec approach. Some results are expected. For example, the cold-start is a known limitation of the IDRec setting.

Comments and questions.
- Chose one acronym between MoRec and CoRec.
- It can be useful to clarify the differences between the learning techniques. Many of them are introduced incrementally and it can lead to some confusions.
- Table 2. Split the table into two parts (one for DSSM and one for SASRec) with one improvement column for each architecture.
- Table 3. I am not sure to understand what the numbers in the column New item/IDRec represent. IDRec cannot learn an item embedding if there is no example available in the dataset. Therefore, is the embedding used for evaluation correspond to the random initialization?
- DT layer. End-to-end training is shown to perform better than two-stage training (but is very expensive to train). One possible explanation is that the obtained representation with the encoder is not well aligned with the recommendation model. The DT layer is only a dense layer in the experiments. Does a better DT layer (using a neutral network) can improve the results of the two-stage approach?


**Summary Of The Paper:**

There are two different paradigms to represent the items in a recommendation setting: (i) IDRec which uses a unique identifier for each item (ii) MoRec which encodes the available content/modalities of the item (text description, images, etc.). IDRec has been dominating the recommender systems literature in the past decade due to its high flexibility. However, a recent trend in the recsys community is to use the recent advances in NLP and CV to build a powerful MoRec model. In particular, one of the promise of this approach is to develop of a general recommender system which can transfer its knowledge from a dataset to another.
In this paper, the authors propose an empirical comparison of the two approaches (IDRec vs MoRec). They compare two different model architectures (two-tower and sequential models) and several encoders (BERT, RoBERTa, Swin, etc.) on three datasets.
The main conclusion of this paper is that MoRec can be highly competitive with IDRec and benefits the recent advances in NLP and CV.



**Summary Of The Review:**

The paper is very pleasant to read and well structured. The authors propose an empirical comparison of two paradigms (IDRec and MoRec). This comparison is useful for the recsys community since there is a recent trend in building a general recommender system. However, there is no technical novelty in the paper.

---

> ### Author Response · Authors · 2022-11-13
> **Reply3: To reviewer jqDE. Regarding novelty.**
>
> Dear reviewer jqDE
>
> Here,**we hope to further clarify our novelty and contributions in this paper.**
>
>
> First, we would like to say, although there is some peer-reviewed literature that proposed new models and claimed SOTA. But unfortunately, none of these formally published papers has done a fair comparison, i.e. comparing their proposed MoRec with a proper IDRec. **However,  whether MoRec can outperform or shake the dominant role of IDRec is $\textcolor{red}{a \ fundamental  \ question}$.** In fact, most exisiting literature used a weak backbone for IDRec or use a bad setting (we listed such papers at the top of this page). Since no published paper has done a fair study for MoRec and IRec, we believe our work is very new and meaningful. Our study and conclusion are very important for guiding future research in the community.
>
> Finally, we want to emphasize our key findings. We believe these findings are novel!
>
> (1) We are the first to find that MoRec with DSSM backbone is not a good choice, falling far short of IDRec's performance.  **No paper has told the community the backbone network should be carefully considered when replacing ID with modality encoder.** We believe this is new and important for researchers in this direction.
>
> (2) We are the first to empirically demonstrate that  MoRec （SASRec +Swin Transformer) can achieve near SOTA (comparable with IDRec) for image recommendation (on the non-cold setting).  This is a very big progress. There is no peer-reviewed MoRec (modality-only Rec) paper for image RS using E2E learning except [1] (still a preprint). Like other literature, [1] did not compare with IDRec and used a 20x smaller dataset than ours.
>
> (3) We are also the first to demonstrate that MoRec with text modality encoder can already outperform IDRec. Although there are a few papers doing E2E text MoRec, no one has clearly demonstrated this.
>
> (4) We are the first to benchmark so many modality encoders for MoRec and found that some improvements made in NLP and CV could be inherited by RS community, this is very important findings. It is not easy to perform such research since for most image modality encoders, MoRec has to be trained with **100x-1000x** longer time and compute. Our findings in so many large datasets are new.
>
>
> (5) We demonstrated both positive and negative facts, e.g. by replacing modality encoder with ever-bigger BERT, MoRec cannot always be improved. By performing a second round of pre-training on the image item encoder, E2E MoRec can be improved.
>
>
> (6) Although some researchers know E2E could perform better than 2-stage training. However, so far there is not even one E2E MoRec that is successfully deployed in a real system according to literature. In this paper, we highlight that the widely used two-stage method will cause **non-ignorable** performance drop. This is what we want to tell the industrial practitioners. We also show that E2E requires nearly 1000x longer training time, and several hundreds of times larger compute. This is not clearly mentioned in previous E2E literature.
>
> (7)We present several other tricks in section 7, some are widely used but do not work, and some are unknown such as using different learning rate for the user and item encoder.
>
> (8) **We will release a large image dataset with raw image features that can be used for E2E training.** To our best knowledge, this is the only image dataset that contains rich image contents, which is vastly different from HM that only includes a single object i.e., clothes and shoes. Images in our Bili look more difficult to understand compared with HM  (see our appendix, very clear). We believe a large-scale high-quality dataset is of great value for this community.   **In fact, we have collected over 2 million users, 150,000 images, and over 50 million interactions ($\textcolor{red}{a \  very \ huge \ dataset}$) for over a year. We promise to make this dataset public if this paper is accepted!  We believe even such a large and unique dataset can be an ICLR paper. (More research work can be inspired)**
>
> **For this paper, it includes empirical study, revisiting study, item encoder benchmark and dataset releasing.** Hope we can convince you to accept it. We have spent up to 100,000$ dollars and over 15 months doing these empirical studies. We believe our paper's contribution could be even more important and valuable than developing a new RS model.**
>
>
>
>
>
> [1] Elsayed et al, End-to-End Image-Based Fashion Recommendation. pre-print 2022

---

### Author Response · Authors · 2022-11-12
**For all reviewers, regarding novelty.(papaer score 8 6 5 5  3)**

Hi all, thanks a lot!

Here,**we want to further clarify our novelty and contributions in this paper.**

First, we hope to clarify  the misunderstanding --- we never claimed the MoRec framework was  a contribution by replacing ID embedding with modality encoder (ResNet, BERT). This is a common way and that is the reason we need to carefully study it. We guess reviewer BBQN might have a misunderstanding.

Second, although there are some  peer-reviewed literature that proposed new models or claimed new SOTA. yet **none of these papers has done a fair or explicit comparison, i.e., comparing the proposed MoRec to a proper IDRec**.  In fact, most literature used $\textcolor{red}{a \ weak \  backbone \ for \ IDRec \ as  \ the   \ baseline}$. However, whether MoRec (modality-only RS) can perform better or on par with IDRec is $\textcolor{red}{a\ fundamental \ question}$, which could potentially change the mainstream research direction. Our findings are very important for guiding new research in the community.  In fact, **ICLR2022,2021 and NIPS had many empirical study papers although such papers do not focus on  technical novelty.**

Finally, we want to clearly show our key findings, we believe they are new!

(1) We are the first to find that MoRec with DSSM backbone is not a good choice, falling far short of IDRec's performance.  **No paper has told the community the backbone network should be carefully considered when replacing ID with modality encoder.** We believe this is new and important for researchers in this direction.

(2) We are the first to empirically demonstrate that  MoRec （SASRec +Swin Transformer) can **achieve near SOTA (comparable with IDRec for the first time) for image recommendation (on the non-cold setting).**  **This is a very important and new finding for modality- or image-only reecommendation**.

(3) We are also the first to clearly demonstrate that MoRec (SASRec+ BERT, using only  text features)  can already outperforms IDRec. No one has provided a fair comparison in peer-reviewed literature. We also show that MoRec with ResNet still cannot outperform IDRec.
These findings are new and could guide new research in this direction.

(4) We are the first to benchmark so many modality encoders for MoRec and found that some advances made in NLP and CV could be inherited by RS community, these are very important findings. It is not easy to perform such research since for most image modality encoders (e.g., ResNet, Swin Transformer), MoRec has to be trained with **100x-1000x** longer time & compute.  **The community needs someone to do such research.**

(5) We demonstrated both positive and negative facts, e.g., by replacing modality encoder with ever-bigger BERT, MoRec cannot always be improved. By performing a second round of pre-training on image item encoder, E2E MoRec cannot be improved. Without experiments, we cannot know them.


(6) Some researchers may know that E2E usually performs better than the 2-stage training. However, so far there is no even one E2E MoRec  that is successfully deployed in a real system according to literature. In this paper, we highlight that the widely used two-stage method will cause **non-negligible performance drop**. This is what we want to tell the industrial practitioners, in particular for image RS (there is no peer-reviewed E2E paper for MoRec (image-only) RS so far). We also show that E2E requires nearly 1000x longer training time, 300x larger compute. This is not explicitly mentioned in previous E2E literature.

(7)We present several other findings in section 7, some are widely used but not work, some are unknown such as using different learning rate for user &  item encoder.


**For this paper, it includes empirical study, revisiting study, item encoder benchmark and dataset releasing.

[1] Elsayed et al, End-to-End Image-Based Fashion Recommendation. pre-print 2022

---

### Decision · Program_Chairs · 2023-01-20

**Decision:**

Reject

**Justification For Why Not Higher Score:**

Just to elaborate a bit more about the points above (I ran out of space): in the case where the objective is to predict interaction, in theory an unconstrained model (IDRec) should do no worse than a constrained model (MoRec) on **optimizing the objective function**. However, there are many other important factors in place, some of which conflict with each other, for example, an important one being the objective function in recommender systems doesn't necessarily correlate with the commonly-used metrics, furthermore the actual optimization might be easier/harder with and without a pre-trained model especially as we move beyond simpler linear models as we do these days. Not to mention the role of popularity which the modality-based approach has always had a difficult time with (e.g., imagine a viral video and many followup copycats which share similar content features but with very different levels of user interactions). Therefore in practice we observe different results in different problems/datasets.

I think this paper will be much stronger coming from this perspective of "here is what is supposed to happen in theory but in practices these are the factors that will make things more complicated and we isolate each of them one-by-one".


**Justification For Why Not Lower Score:**

N/A

**Metareview: Summary, Strengths And Weaknesses:**

Summary: It is common in recommender systems that an item is represented as a unique identity (ID) and subsequently turned into an embedding vector as free parameters which are learned from scratch. On the other hand, this paper seeks to answer the following question: what if we use the content (modality) of the item to construct this embedding vector via the pre-trained state-of-the-art architectures in their respective domains (e.g., BERT for text and ResNet for images)? The paper conducts extensive empirical studies to demonstrate that modality-based (MoRec) recommender systems perform comparably with the ID-based (IDRec) recommender systems, and sometimes even outperform IDRec. The paper also dives deeper into various aspects of the comparison, e.g., comparing end-to-end training with two-stage training.

Strengths: All the reviewers agree that this paper conducted extensive experiments and the problem it aims to address is certainly an important one. The paper is overall well-written.

Weaknesses: A major weakness raised by the majority of the reviewers is the lack of novelty, to which the authors disagree and argue that "the dominant role of IDRec" in the recommender systems community should be questioned and this paper did exactly that. I read the paper myself this past week and I would like to offer some historical context from my observation/experience, some of which the authors seem to have a bit of misunderstanding.

It is true that historically IDRec is preferred over context-based recommender systems, but here the "context-based recommender systems" refers to something rather different from what the authors seem to suggest. In the earlier days, recommender systems/collaborative filtering were mostly based on similarity. Here the similarity can be defined by interaction/ratings (i.e., two items are considered "similar" if they are rated by the same group of users), e.g., the early neighborhood-based method in Sarwar et al. "Item-based collaborative filtering recommendation algorithms" (2001). And of course the similarity can also be defined by the content of the item (i.e., two items are considered "similar" if they live close in the content feature space). What people have found is that the similarity defined by interaction/ratings is much more useful than the similarity defined by the content. And this is exactly why the early "content-based recommender systems" did not perform very well -- it did not take the interactions into account and was purely based on the content of the items, while in reality, the interactions convey much more information than the content alone.

However, in this paper, the MoRec uses the content of the item to construct the embeddings which are later used to predict interactions in the objective function, which is exactly the learning people have discovered over the years. If we think in a standard matrix factorization setting (for simplicity) which is a linear case of the two-tower model used in this paper, for user $u$ we have a user representation $\theta_u$ and for item $i$ we have an item representation $\beta_i$, let's say we optimize some loss function $l(y_{ui}, \theta_u^\top \beta_i)$ where $y_{ui}$ is the target we want to predict (e.g., 1 for click and 0 otherwise) and $l$ can be binary cross-entropy, BPR, or whatever. Here $\theta_u$ and $\beta_i$ are free parameters. If now we replace item factor $\beta_i$ with $f(x_i)$ where $f$ is a modality-encoder as called in this paper and $x_i$ is the raw item feature. Our loss function becomes $l(y_{ui}, \theta_u^\top f(x_i))$. Which one is better? In theory, there should exist an optimal item factor $\beta_i^*$ and if the modality-encoder $f$ is expressive enough it is certainly possible to learn $f(x_i) = \beta_i^*$. Of course there are many important factors that I didn't consider (e.g., an important one being the loss function in recommender systems doesn't necessarily correlate with the commonly-used metrics, furthermore the actual optimization might be easier/harder with and without a pre-trained model especially as we move beyond simpler linear models as we do these days), but I think it is clear that in this setting where the objective is to predict interaction, there doesn't exist a clear winner and it will likely be dataset- and problem-dependent.

In fact, precisely for this reason, there has been work dating back to a decade ago which aims to combine both the content and interaction in a single model to take the best of both worlds (one such example is $l(y_{ui}, \theta_u^\top (\beta_i + f(x_i)))$, see Wang & Blei, "Collaborative Topic Modeling for Recommending Scientific Articles" 2011, which was also awarded the test of time award in 2021, and many followup work along this line).

This paper has certainly done some important empirical studies. However, I hope the authors can better understand the research landscape and more carefully articulate some of the statements.

---

> ### Author Response · Authors · 2023-02-23
> **The paper has been accepted in SIGIR2023, public online in 2022/09 by Openreview**
>
> We respectfully disagree with the comments made by some reviewers in ICLR, disappointing!
>
> Our camera-ready version is in: **https://arxiv.org/pdf/2303.13835.pdf**
>
>
> Review Feedback from SIGIR perspective track:
>
> **"This is an interesting and original analysis and discussion. While the solution is not optimal, it should spark additional research on several open problems mentioned in this paper, and thus has potentially high impact."**
>
>
> **"I think it addresses an interesting and promising big direction. It will help foster discussion along this line of research in IR."**